# The folded spin-$1/2$ XXZ model: II. Thermodynamics and hydrodynamics with a minimal set of charges

**Lenart Zadnik, Kemal Bidzhiev and Maurizio Fagotti⋆**

Université Paris-Saclay, CNRS, LPTMS, 91405, Orsay, France

⋆ maurizio.fagotti@universite-paris-saclay.fr

## Abstract

We study the (dual) folded spin-1/2 XXZ model in the thermodynamic limit. We focus, in particular, on a class of "local" macrostates that includes Gibbs ensembles. We develop a thermodynamic Bethe Ansatz description and work out generalised hydrodynamics at the leading order. Remarkably, in the ballistic scaling limit the junction of two local macrostates results in a discontinuity in the profile of essentially *any* local observable.


# 1 Introduction

This is the second part of our investigation into the large-anisotropy limit of the Heisenberg spin-1/2 XXZ model. In the first part [1] we defined the "folded picture" as an asymptotic formulation of quantum mechanics of many-body systems that are described by Hamiltonians with a large coupling constant. In this picture the fast oscillatory part of the dynamics is attached to the operators, whereas the state evolves slowly in time through an effective "folded Hamiltonian". As an example we considered the folded XXZ model, which reveals a structure otherwise hidden in the standard Bethe Ansatz solution [2] of the anisotropic Heisenberg magnet. Our solution of the folded XXZ model is peculiar as it diagonalises the folded Hamiltonian, but not the total spin $\mathbf{S}^z$ in the direction of the anisotropy. This is unveiled by a duality transformation that maps the folded Hamiltonian into a block-diagonal local operator and $\mathbf{S}^z$ into a block off-diagonal pseudolocal one.

The present manuscript is a detailed account of the thermodynamic properties of the dual folded XXZ model, both in and out of equilibrium. Up to boundary terms, the Hamiltonian that we consider reads as $J \sum_\ell (\sigma^x_{\ell-1} \sigma^x_{\ell+1} + \sigma^y_{\ell-1} \sigma^y_{\ell+1}) \frac{1-\sigma^z_\ell}{2}$, where $\sigma^\alpha_\ell$ are the Pauli matrices. It describes a special point of the two-component Bariev model [3], which is solvable with a nested Bethe Ansatz technique. In order to exploit the symmetries of that special point, we will however use the results of the non-nested Bethe Ansatz method of Ref. [1]. We first develop a thermodynamic Bethe Ansatz that describes the infinite chain in thermal states as well as in generalised Gibbs ensembles [4] constructed with the local conservation laws and a special pseudolocal charge. The predictions of the thermodynamic Bethe Ansatz are checked against numerical data from DMRG algorithms.

The macrostates represented by generalised Gibbs ensembles are the key ingredients of the generalised hydrodynamic theory. Introduced in Refs [5,6], and recently experimentally corroborated in cold-atom setups [7], this mesoscopic description of the dynamics in integrable systems allows one to treat the evolution of inhomogeneous states on large space-time scales, and also in presence of inhomogeneous space- and time-dependent interactions [8,9]. Generalised hydrodynamics is based on the assumption that the so-called root densities, which characterise the local properties of stationary states, can be promoted into functions of space and time. The validity of such an assumption is hard to establish rigorously [10,11], and it is typically justified by comparing the predictions with state-of-the-art numerical simulations. Provided that the assumption holds, the state can be coarse-grained into fluid cells locally equivalent to macrostates, which can be accessed through the thermodynamic Bethe Ansatz.

Within this framework, we investigate the transport phenomena that emerge after two grand canonical ensembles are joined together and left to evolve in time under the folded Hamiltonian. Remarkably, due to the rich particle content of the folded XXZ model, a local macrostate is not completely determined by a root density, and an additional independent (pseudolocal) thermodynamic variable is necessary for describing the junction of two local macrostates. We show that this special feature results in discontinuous ballistic-scale profiles of all local charges, independently of how the initial grand canonical ensembles are prepared.

In the rest of the introduction we recapitulate the results of Ref. [1] necessary to understand the present work.

## 1.1 The asymptotic folded picture

When a Hamiltonian has a large coupling constant, it is possible to derive asymptotic expansions of the time evolution operator in the strong coupling limit and at fixed time. Starting from Ref. [12], explicit results have been obtained for Hamiltonians $\mathbf{H}(\kappa)$ of the form

$$\mathbf{H}(\kappa) = \mathbf{H}_F + \sum_{m=1}^{q}(\mathbf{F}_m + \mathbf{F}_m^\dagger) + \kappa^{-1}\mathbf{H}_I\,, \tag{1}$$

where $q$ is a finite integer and

$$[\mathbf{H}_I, \mathbf{H}_F] = 0\,, \qquad [\mathbf{H}_I, \mathbf{F}_m] = mJ\mathbf{F}_m\,. \tag{2}$$

In Ref. [1] we have recast the strong coupling expansion $\kappa \to 0$ of such models into a formulation termed "folded picture", in which operators and state time evolve according to

$$\mathbf{O}_F(t) := e^{i\kappa\mathbf{B}_{z_t}(\kappa)}e^{i\kappa^{-1}\mathbf{H}_I t}\mathbf{O}e^{-i\kappa^{-1}\mathbf{H}_I t}e^{-i\kappa\mathbf{B}_{z_t}(\kappa)}\,, \qquad |\Psi(t)\rangle_F := e^{-i\mathbf{H}_F(\kappa)t}e^{i\kappa\mathbf{B}_1(\kappa)}|\Psi(0)\rangle\,, \tag{3}$$

where $z_t = e^{i\kappa^{-1}Jt}$. Asymptotic expansions of $\mathbf{H}_F(\kappa)$ and $\mathbf{B}_z(\kappa)$ in the limit of small $\kappa$ are reported in Ref. [1]. As long as $Jt \ll \kappa^{-1}$, the folded picture can be used at the leading order, where the state time evolves with $\mathbf{H}_F(0) \equiv \mathbf{H}_F$. This was used for example in Ref. [13] to explain the very slow restoration of one-site shift invariance in XY and XXZ models. Note that, contrary to the standard interaction picture, in the folded picture it is the state rather than the operator that evolves with a time-independent Hamiltonian; furthermore, both state and operators undergo an additional unitary transformation generated by $\mathbf{B}_1(\kappa)$.

Following this perspective, the present paper investigates time evolution under $\mathbf{H}(\kappa)$ in the asymptotic limit $1 \ll Jt \ll \kappa^{-1} \ll L \to \infty$. In fact, we will assume the relation between $\mathbf{H}(\kappa)$ and $\mathbf{H}_F$ as understood, and interpret $\mathbf{H}_F$ as the Hamiltonian of a new model, which we refer to as "folded" model. We are using this nomenclature because, in the limit $\kappa \to 0$, the spectrum of $\mathbf{H}(\kappa)$ turns out to be equal to the spectrum of $\mathbf{H}_F$, modulo a typical energy, inversely proportional to $\kappa$.

## 1.2 The folded XXZ model and its dual

The XXZ spin-1/2 chain is described by the Hamiltonian

$$\mathbf{H} = J\sum_\ell \sigma_\ell^x\sigma_{\ell+1}^x + \sigma_\ell^y\sigma_{\ell+1}^y + \Delta\sigma_\ell^z\sigma_{\ell+1}^z\,. \tag{4}$$

We are interested in the limit of large anisotropy $\Delta$. In the folded picture we identify $\kappa$ with $(4\Delta)^{-1}$, and the operator $\mathbf{H}_F$ is given by[1]

$$\mathbf{H}_F = J\sum_\ell \frac{1 + \sigma_{\ell-1}^z\sigma_{\ell+2}^z}{2}(\sigma_\ell^x\sigma_{\ell+1}^x + \sigma_\ell^y\sigma_{\ell+1}^y)\,. \tag{5}$$

---

[1]$\mathbf{H}_F$ can be obtained, e.g., by factoring $e^{-i\kappa^{-1}\mathbf{H}_I t}$ out of the full XXZ time evolution, keeping only the leading order in $\kappa$ of the remaining part $e^{i\kappa^{-1}\mathbf{H}_I t}e^{-i\mathbf{H}(\kappa)t}$.

In the following we provide a brief overview of the diagonalisation of $\mathbf{H}_F$ that we worked out in the first part of our work [1].

The preliminary step is a duality transformation mapping $\mathbf{H}_F$ into a local operator with a density that has a range of three sites. The transformation reads

$$
\boldsymbol{\sigma}_\ell^x \mapsto \begin{cases} -\boldsymbol{\sigma}_1^y \prod_{j=2}^{L-1} \boldsymbol{\sigma}_j^z \boldsymbol{\sigma}_L^y, & \ell = 1, \\ \boldsymbol{\sigma}_{\ell-1}^x \boldsymbol{\sigma}_\ell^x, & \ell > 1, \end{cases}
$$

$$
\boldsymbol{\sigma}_\ell^y \mapsto \begin{cases} \boldsymbol{\sigma}_1^x, & \ell = 1, \\ \boldsymbol{\sigma}_{\ell-1}^x \boldsymbol{\sigma}_\ell^y \prod_{j=\ell+1}^{L-1} \boldsymbol{\sigma}_j^z \boldsymbol{\sigma}_L^y, & 1 < \ell < L, \quad \boldsymbol{\sigma}_\ell^z \mapsto \prod_{j=\ell}^{L-1} \boldsymbol{\sigma}_j^z \boldsymbol{\sigma}_L^y, \\ -\boldsymbol{\sigma}_{L-1}^x \boldsymbol{\sigma}_L^z, & \ell = L, \end{cases} \tag{6}
$$

where $L$ is the chain's length. Under the transformation (6), the folded Hamiltonian with periodic boundary conditions ($\boldsymbol{\sigma}_{L+n}^\alpha = \boldsymbol{\sigma}_n^\alpha$, for $\alpha \in \{x,y,z\}$) is mapped into the following operator

$$
\tilde{\mathbf{H}}_F = \frac{1+\mathbf{\Pi}^z}{2} \tilde{\mathbf{H}}_F^0 \frac{1+\mathbf{\Pi}^z}{2} + \frac{1-\mathbf{\Pi}^z}{2} \boldsymbol{\sigma}_L^x \tilde{\mathbf{H}}_F^1 \boldsymbol{\sigma}_L^x \frac{1-\mathbf{\Pi}^z}{2}, \tag{7}
$$

where $\mathbf{\Pi}^z = \prod_{\ell=1}^L \boldsymbol{\sigma}_\ell^z$, and

$$
\tilde{\mathbf{H}}_F^\eta = J \sum_{\substack{\ell=1 \\ \sigma_{L+n}^{x,y} = (-1)^\eta \sigma_n^{x,y}}}^L (\boldsymbol{\sigma}_{\ell-1}^x \boldsymbol{\sigma}_{\ell+1}^x + \boldsymbol{\sigma}_{\ell-1}^y \boldsymbol{\sigma}_{\ell+1}^y) \frac{1-\boldsymbol{\sigma}_\ell^z}{2}. \tag{8}
$$

The eigenstates of $\tilde{\mathbf{H}}_F^\eta$ with $\mathbf{\Pi}^z = 1$ are mapped into eigenstates of $\tilde{\mathbf{H}}_F$ either trivially, or by the unitary transformation $\boldsymbol{\sigma}_L^x$, depending on whether $\eta = 0$ or $\eta = 1$. It is therefore convenient to focus on $\tilde{\mathbf{H}}_F^\eta$. As already mentioned, the latter Hamiltonian describes a strong repulsion limit of the two-component Bariev model [3]. In our previous work [1] we circumvented the more general solution (based on nested Bethe Ansatz) by exploiting the additional symmetries emerging in such a strong repulsive regime. This allowed us to solve the Bethe equations in terms of closed-form relations between momenta and quantum numbers, which will be summarised in the next section.

## 1.3 Coordinate Bethe Ansatz

A basis of eigenstates of $\tilde{\mathbf{H}}_F^\eta$ can be constructed within a coordinate Bethe Ansatz starting from the reference state

$$
|\text{vac}\rangle = |\downarrow\downarrow \ldots \downarrow\rangle. \tag{9}
$$

For the sake of simplicity we assume that $L$ is even; the reader can find some information about the odd case in Ref. [1]. We label the positions of the $N$ spins up in an eigenstate by $2\ell_j' - b_j$ ($j = 1, 2, \ldots, N$), where $\ell_j' \in \{1, 2, \ldots, L/2\}$ indicates the *macrosite*, i.e., a pair of neighbouring spin sites, and $b_j \in \{0, 1\}$ the parity of the spin's position. Within this convention the set $B_N = \{(b_1, \ldots, b_N)\}_c$ of cyclic permutations of the sequence $(b_1, \ldots, b_N)$ is preserved by the Hamiltonian. In the following we will refer to the information encoded in $B_N$ as that pertaining to the "configuration space". The eigenstates are characterised by $N$ momenta (rapidities) $\{p_\ell\}_{\ell=1}^N$, each one associated with a spin up in an even ($b = 0$) or odd ($b = 1$) position. Interactions are characterised by the two-body scattering matrix

$$
S_{b_1,b_2}\binom{p_1,p_2}{p_2,p_1} = -1 + b_1(1-b_2)\left(1 - e^{i(p_1-p_2)}\right). \tag{10}
$$

The energy of the state $|p_1, \ldots, p_N\rangle_{B_N}$, specified by the momenta and the configuration, is given by

$$E = 4J \sum_{\ell=1}^{N} \cos p_\ell \,. \tag{11}$$

The momenta $p_1, \ldots, p_N$ solve Bethe equations that can be integrated explicitly, as shown in Ref. [1]. The solution reads

$$p_\ell = \frac{2\pi\left(I_\ell + \frac{\varphi}{2\pi}\right) + \frac{M}{N}P}{\frac{L}{2} + M}\,, \tag{12}$$

where the total momentum and the shift of quantum numbers are defined as

$$P = \frac{4\pi}{L} \sum_{\ell=1}^{N}\left(I_\ell + \frac{\varphi}{2\pi}\right), \qquad \frac{\varphi}{2\pi} = \frac{\eta + g - 1}{2} + \frac{gI_0}{N}\,, \tag{13}$$

respectively. The integer quantum numbers $I_\ell$ satisfy

$$0 \le I_1 < I_2 < \cdots < I_N < \frac{L}{2} + M\,, \qquad 0 \le I_0 < \frac{N}{g}\,, \tag{14}$$

$N/g = \min\{m|b_{n+m} = b_n, \forall n\}$ denoting the size of the unit cell in the sequence $(b_1, \ldots, b_N)$, and $M = \sum_{j=1}^{N} b_j(1 - b_{j+1})$, with $b_{N+1} = b_1$ the number of subsequences $(1,0)$.

We point out that some configurations of integers within the domain specified by Eq. (14) give rise to the same set of momenta. This subtlety, however, does not affect the thermodynamic limit, so we will not be more specific about it; the interested reader can find more details in Ref. [1]. Since the momenta are invariant under the transformation $I_\ell \to I_\ell - n$, $\varphi \to \varphi + 2\pi n$, with $n \in \mathbb{Z}$, the parameter $\varphi$ can be defined modulo $2\pi$.

An alternative parametrisation of the solution to the Bethe equations is through the rational quantum numbers $J_\ell = I_\ell + \frac{\varphi}{2\pi}$, in terms of which we have

$$p_\ell = \frac{2\pi J_\ell + \frac{M}{N}P}{\frac{L}{2} + M}\,, \qquad P = \frac{4\pi}{L} \sum_{\ell=1}^{N} J_\ell\,. \tag{15}$$

We emphasise that the rational quantum numbers lie in the affine lattice $\mathbb{Z} + \frac{\varphi}{2\pi}$, whose shift, with respect to integers, depends on the state itself. Importantly, for large $N$ almost all the states have $g = 1$ [1]. In light of this, we will refer to them as *generic states*.

## 1.4 Conservation laws

In the first part of our work [1] we have shown that, in finite chains, the momenta of the Bethe Ansatz characterise the eigenvalues of an "extended"[2] family of local conservation laws. Among them, we have identified two charges that are diagonal in the standard $\boldsymbol{\sigma}^z$ basis: $\mathbf{S}^z = \frac{1}{2} \sum_\ell \sigma_\ell^z$ and

$$\mathbf{M} = \sum_{\ell'=1}^{\frac{L}{2}} \sum_{n'=0}^{\frac{L}{2}-1} \frac{1 + \sigma_{2\ell'-1}^z}{2} \left( \prod_{j=2\ell'}^{2\ell'-1+2n'} \frac{1 - \sigma_j^z}{2} \right) \frac{1 + \sigma_{2\ell'+2n'}^z}{2}\,, \tag{16}$$

the latter operator's eigenvalue being equal to the parameter $M$ in the Bethe equations (12). The remaining charges of the family are not diagonal and can be organized into two sequences:

---

[2]The extension comes from allowing the index of the charge, which, up to a given value, represents its range, to be arbitrarily large.

$\mathbf{Q}_n^{\pm} = \sum_{\ell=1}^{L} \mathbf{q}_{n,\ell}^{\pm}$. The local density $\mathbf{q}_{n,\ell}^{\pm}$ can be defined so as to be supported on $2n + 1$ neighbouring sites and, by convention, $\mathbf{Q}_1^+ = \tilde{\mathbf{H}}_F^{\eta}$. The expectation value of any charge $\mathbf{Q}_n^{\pm}$ in the Bethe state can be written as

$$Q = \langle \mathbf{Q} \rangle - \langle \mathrm{vac} | \mathbf{Q} | \mathrm{vac} \rangle = \sum_{\ell=1}^{N} q(p_\ell), \tag{17}$$

where the sum runs over the momenta. Here, $q(p)$ denotes the one-particle eigenvalue of the charge, for example, $q_1^+(p) = 4J \cos p = E(p)$ is the one-particle energy. In Eq. (17) we subtracted the vacuum expectation value of the charge, since the reference state $|\mathrm{vac}\rangle$ has no momenta, i.e., the right-hand side of the equation is zero in that case.

In general, the single-particle eigenvalues of the charges in the integrable hierarchy read

$$q_n^+(p) = 4J \cos(np), \qquad q_n^-(p) = 4J \sin(np), \tag{18}$$

and notably span the Fourier basis in the space of functions of the momentum. As an example, we state the local densities of the first few conservation laws, $\mathbf{Q}_1^{\pm}$ and $\mathbf{Q}_2^+$:

$$
\begin{aligned}
\mathbf{q}_{1,\ell}^+ &= \frac{J}{2} \mathbf{K}_{\ell,\ell+2} (1 - \boldsymbol{\sigma}_{\ell+1}^z), \\
\mathbf{q}_{1,\ell}^- &= -\frac{J}{2} \mathbf{D}_{\ell,\ell+2} (1 - \boldsymbol{\sigma}_{\ell+1}^z), \\
\mathbf{q}_{2,\ell}^+ &= -\frac{J}{4} \Big[ \mathbf{K}_{\ell+1,\ell+4} \mathbf{K}_{\ell+2,\ell+3} - \mathbf{D}_{\ell+1,\ell+4} \mathbf{D}_{\ell+2,\ell+3} + (1 - \boldsymbol{\sigma}_{\ell+1}^z) \boldsymbol{\sigma}_{\ell+2}^z (1 - \boldsymbol{\sigma}_{\ell+3}^z) \mathbf{K}_{\ell,\ell+4} \Big], \\
\mathbf{q}_{2,\ell}^- &= \frac{J}{4} \Big[ \mathbf{D}_{\ell+1,\ell+4} \mathbf{K}_{\ell+2,\ell+3} + \mathbf{K}_{\ell+1,\ell+4} \mathbf{D}_{\ell+2,\ell+3} + (1 - \boldsymbol{\sigma}_{\ell+1}^z) \boldsymbol{\sigma}_{\ell+2}^z (1 - \boldsymbol{\sigma}_{\ell+3}^z) \mathbf{D}_{\ell,\ell+4} \Big],
\end{aligned}
\tag{19}
$$

where $\mathbf{K}_{n,m} = \boldsymbol{\sigma}_n^x \boldsymbol{\sigma}_m^x + \boldsymbol{\sigma}_n^y \boldsymbol{\sigma}_m^y$, $\mathbf{D}_{n,m} = \boldsymbol{\sigma}_n^x \boldsymbol{\sigma}_m^y - \boldsymbol{\sigma}_n^y \boldsymbol{\sigma}_m^x$. For any fixed $n$, the conserved charges $\mathbf{Q}_n^{\pm}$ are connected by the adjoint action of $\mathbf{L} = \sum_{\ell=1}^{L} \ell\, \boldsymbol{\sigma}_\ell^z$, namely, $i[\mathbf{L}, \mathbf{Q}_n^{\pm}] = \mathbf{Q}_n^{\mp}$. In integrable spin chains one usually refers to such an operator as a "boost" or "ladder" operator if it allows the reconstruction of the entire hierarchy of local conserved quantities from a single charge. In our case it is evidently not so. Moreover, there seems to be no local boost operator of this form, by means of which one could translate between conserved charges with different $n$. This is consistent with the fact that the energy current (its explicit form is provided in Section 4.2) is not conserved [14], as well as with the fact that the $R$-matrix that constitutes the Algebraic Bethe Ansatz for the two-component Bariev model is not of the difference form [15].

We point out that the family of local charges with local densities (19) is not complete, and we can even exhibit a local charge that does not belong to the family: the staggered spin along the $z$-axis

$$\mathbf{S}_{\mathrm{st}}^z = \frac{1}{2} \sum_{\ell=1}^{L} (-1)^\ell \boldsymbol{\sigma}_\ell^z. \tag{20}$$

We will denote its expectation value by $S_{\mathrm{st}}^z = \frac{L}{2} m_{\mathrm{st}}^z$ and refer to it as staggered magnetisation. The staggered spin along the $z$-axis is one of the diagonal operators that commute with the Hamiltonian and span the configuration space. The common property of such diagonal charges is that their eigenvalues are functionals of the configuration $B_N$.

## Summary

**Section 2** considers the thermodynamic limit of the Bethe equations and develops a (generalised) thermodynamic Bethe Ansatz description. Thermal states are analysed in detail, including low-temperature and high-temperature asymptotic expansions.

**Section 3** proposes a definition of elementary quasiparticles (excitations) and works out their dressed charges.

**Section 4** develops generalised hydrodynamics (GHD) in the dual folded Hamiltonian for a class of states that are characterised by a minimal set of charges.

**Section 5** uses GHD to investigate time evolution after two (local) macrostates have been joined together. It is shown that the profiles of essentially all local observables remain discontinuous in the ballistic scaling limit.

## 2 Thermodynamic Bethe Ansatz

The thermodynamic Bethe Ansatz (TBA) was originally developed by Yang and Yang for the one-dimensional Bose gas with Dirac-delta repulsive interactions [16]. In this section we will follow their approach with few minimal changes necessary for accommodating the TBA in the richer structure of the folded XXZ Hamiltonian. We note that an alternative, albeit more complicated, approach towards TBA exists for our model. It can be obtained as a particular limit of the nested Bethe Ansatz solution of the more general multi-component Bariev model [17].

### 2.1 Root densities and expectation values of charges

We start by noting that the solution (12) to the Bethe equations can be recast into the standard form

$$\frac{L}{2}h(p_\ell) = 2\pi J_\ell, \qquad h(p) := p + \frac{2M}{LN}\sum_{j=1}^{N}(p - p_j), \qquad \ell = 1, 2, \ldots, N.\tag{21}$$

Here we have introduced the monotonous counting function $h(p)$: $\partial_p h(p) = 1 + \frac{\mu}{\xi} > 0$, where we denoted $\mu = M/N$ and $\xi = \frac{L}{2N}$. Whenever evaluated in the momentum that solves the Bethe equations, the value of $\frac{L}{4\pi}h(p)$ falls into the affine lattice $\mathbb{Z} + \frac{g}{N}I_0 + \frac{\eta+g-1}{2}$ populated by quantum numbers $J_\ell$. Bethe equations (21), in particular the prefactor $L/2$ in front of $p_\ell$, manifest that our momentum generates translations for two sites on the spin chain.

The counting function associates certain elements of the affine lattice $\mathbb{Z} + \frac{g}{N}I_0 + \frac{\eta+g-1}{2}$ to momenta that form a particular solution of Bethe equations. Specifically, a Bethe state with quantum numbers $\{J_1, J_2, \ldots, J_N\}$ contains momenta $\{p_1, p_2, \ldots, p_N\}$ that solve $\frac{L}{4\pi}h(p_\ell) = J_\ell$. If, for instance, $J_2 + 1$ is not among the quantum numbers $\{J_1, J_2, \ldots, J_N\}$, the momentum $p$, for which $\frac{L}{4\pi}h(p) = J_2 + 1$, does not belong to the set $\{p_1, p_2, \ldots, p_N\}$ of momenta in the Bethe state. We refer to the vacant quantum number $J_2 + 1$ as a hole, in the sense that adding that quantum number corresponds to an elementary excitation – see Section 3. In the thermodynamic limit (TD) $N, M, L \to \infty$, with fixed ratios $\mu = M/N$ and $\xi = \frac{L}{2N}$, the momenta characterising the excited states become densely distributed. Then, $\frac{L}{4\pi}\mathrm{d}h(p)$ yields the number of vacancies (both, particles and holes) in the infinitesimal interval $[p, p + \mathrm{d}p) \subset [-\pi, \pi)$. Defining the root density $\rho$ and the density of holes $\rho_h$ as

$$\frac{L}{2}\rho(p)\mathrm{d}p = \text{number of particles with momentum in } [p, p + \mathrm{d}p),$$
$$\frac{L}{2}\rho_h(p)\mathrm{d}p = \text{number of holes with momentum in } [p, p + \mathrm{d}p),\tag{22}$$

we obtain $\mathrm{d}h(p) = 2\pi[\rho(p) + \rho_h(p)]\mathrm{d}p$. The derivative of the counting function is thus proportional to the total density of vacancies

$$\partial_p h(p) = 1 + \frac{\mu}{\xi} = 2\pi\rho_t,\tag{23}$$

which is notably independent of the momentum. From Eqs (17) and (22) it then follows

$$\frac{2}{L}Q \xrightarrow{\text{TD}} \int_{-\pi}^{\pi} \mathrm{d}p \rho(p) q(p),  \tag{24}$$

which is the standard way to express the charge densities as functionals of the root density in the thermodynamic limit.

We are now in a position to recast Eq. (23) in the form of a standard TBA equation. Indeed, the macrosite density of particles is given by

$$\xi^{-1} = \frac{2N}{L} \xrightarrow{\text{TD}} \int_{-\pi}^{\pi} \mathrm{d}p \rho(p),  \tag{25}$$

whence we have – *cf.* Eq. (23):

$$\boxed{\rho_{\mathrm{t}} = \frac{1}{2\pi} + \frac{\mu}{2\pi} \int_{-\pi}^{\pi} \mathrm{d}p \rho(p).}  \tag{26}$$

It is also customary to define the filling function as the density of occupied vacancies, namely, $n(p) = \rho(p)/\rho_{\mathrm{t}}$, which clearly satisfies $0 \leq n(p) \leq 1$. We then have the analogous equation

$$\boxed{\rho_{\mathrm{t}} = \frac{1}{2\pi - \mu \int_{-\pi}^{\pi} \mathrm{d}p\, n(p)}.}  \tag{27}$$

## 2.2 Macrostates and Yang-Yang entropy

In the thermodynamic limit different states can share the same local properties and are usually said to be "locally equivalent". The concept of state is then replaced by the concept of macrostate, which represents the set of all locally indistinguishable states. It is usually understood that macrostates are stationary, and this will be assumed in this section.

We consider first the macrostates characterised by the strictly local integrals of motion; in order to distinguish them from more general macrostates, we will refer to them as "local macrostates". A local macrostate corresponds to a set of microstates (representative states) that, by construction, are simultaneous eigenstates of all quasilocal conservation laws [18]. The expectation value of a quasilocal charge takes, however, the most probable value at fixed local integrals of motion. In particular, as long as the Hamiltonian is local, the equilibrium properties at finite temperature are described by local macrostates. The situation is more complicated when the system is out of equilibrium. For example, local macrostates in the XXZ model were originally conjectured to describe the late-time properties after quantum quenches [19, 20]. The failure of such an assumption was reported in Refs [21, 22], and Ref. [23] later pointed out that the model possesses additional quasilocal charges (i.e., conserved operators with exponentially localised densities) that constrain the dynamics to a greater extent. It soon became evident that the state at late times becomes equivalent to a "quasilocal macrostate", characterised by all quasilocal integrals of motion [24]. In our specific case the stationary properties in the particle space are completely determined by the local charges and a single quasilocal one, **M**. The rest of the charges, diagonal and generally quasilocal, characterise solely the configuration space, i.e., their eigenvalues are functionals of the configuration $B_N$ of a Bethe state.

### 2.2.1 Local macrostates

Except for the staggered magnetisation $\langle \mathbf{S}_{\mathrm{st}}^z \rangle \equiv S_{\mathrm{st}}^z = \frac{L}{2} m_{\mathrm{st}}^z$, all local integrals of motion that we identified are completely determined by the momenta (rapidities), which depend on the configuration through $M$. An important representation of a local macrostate is the (generalised) canonical state, which is described by a density matrix of the form

$$\boldsymbol{\rho} = \frac{1}{Z} \exp\left( h \mathbf{S}_{\mathrm{st}}^z - \sum_n \lambda_n^+ \mathbf{Q}_n^+ - \sum_n \lambda_n^- \mathbf{Q}_n^- - \lambda_0 \mathbf{S}^z \right). \tag{28}$$

Such a state minimises the "generalised free energy", namely, the functional[3]

$$\mathfrak{f}_{\mathrm{loc}}[\boldsymbol{\rho}] = \frac{2}{L} \mathrm{tr}[\boldsymbol{\rho} \log \boldsymbol{\rho}] - \frac{2}{L} \mathrm{tr}\left[ \left( h \mathbf{S}_{\mathrm{st}}^z - \sum_n \lambda_n^+ \mathbf{Q}_n^+ - \sum_n \lambda_n^- \mathbf{Q}_n^- - \lambda_0 \mathbf{S}^z \right) \boldsymbol{\rho} \right], \tag{29}$$

under a generic trace-preserving variation of $\boldsymbol{\rho}$. A local macrostate of the folded XXZ model is characterised by three thermodynamic quantities: $\mu$, $m_{\mathrm{st}}^z$, and the root density $\rho(p)$. In turn, the variation of $\boldsymbol{\rho}$ is realised by an arbitrary variation of $\mu$, $m_{\mathrm{st}}^z$, and $\rho(p)$ in the functional $\mathfrak{f}_{\mathrm{loc}}[\mu, m_{\mathrm{st}}^z, \rho]$, which represents the thermodynamic limit of $\mathfrak{f}_{\mathrm{loc}}[\boldsymbol{\rho}]$. The thermodynamic limit of the second term on the right hand side of Eq. (29) has already been worked out – it is an expectation value of the form (24). The first term, on the other hand, is the entropy of the state per macrosite and is proportional to the logarithm of the number of microstates associated with the particular macrostate [16]. In our specific case, it consists of two terms: the entropy in the configuration space and the entropy in the particle space; both are computed below.

**Entropy in the configuration space.** The number of generic configurations $B_N$ (for the definition, see Section 1.3) that share the same set of momenta $\{p_\ell\}_{\ell=1}^N$ has been computed in Ref. [1] and behaves asymptotically as $\frac{2}{N}\binom{N}{2M}$. The corresponding entropy per macrosite is therefore

$$s_B[\mu, \rho] \sim \frac{2}{L} \log \frac{2}{N}\binom{N}{2M} \xrightarrow{\mathrm{TD}} \xi^{-1} H(2\mu), \tag{30}$$

where $H(p) = -p \log p - (1-p) \log(1-p)$ is the binary entropy function. An analogous calculation gives the entropy at fixed staggered magnetisation $S_{\mathrm{st}}^z = \frac{L}{2} m_{\mathrm{st}}^z$ [1],

$$s_B[\mu, m_{\mathrm{st}}^z, \rho] \sim \frac{2}{L} \log \frac{2}{N}\binom{\frac{N+S_{\mathrm{st}}^z}{2}}{M}\binom{\frac{N-S_{\mathrm{st}}^z}{2}}{M} \xrightarrow{\mathrm{TD}}$$
$$\xrightarrow{\mathrm{TD}} \xi^{-1}\left[ \frac{1-\xi m_{\mathrm{st}}^z}{2} H\left(\frac{2\mu}{1-\xi m_{\mathrm{st}}^z}\right) + \frac{1+\xi m_{\mathrm{st}}^z}{2} H\left(\frac{2\mu}{1+\xi m_{\mathrm{st}}^z}\right) \right], \quad (31)$$

which reduces to the previous expression for $m_{\mathrm{st}}^z = 0$.

**Entropy in the particle space.** In the thermodynamic limit the distribution of momenta is encoded in the root density $\rho(p)$. Following Ref. [16], the entropy associated with the number of microstates with the same root density is

$$s_{\{p\}}[\mu, m_{\mathrm{st}}^z, \rho] \sim \frac{2}{L} \log \prod_p \binom{\frac{L}{2}\rho_{\mathrm{t}} \mathrm{d}p}{\frac{L}{2}\rho(p)\mathrm{d}p} \xrightarrow{\mathrm{TD}} \int_{-\pi}^{\pi} \mathrm{d}p\, \rho_t H\left(\frac{\rho(p)}{\rho_{\mathrm{t}}}\right), \tag{32}$$

where index $p$ in the product runs over the momenta representing the centres of $N$ cells of width $\mathrm{d}p$; they are coarse-grained momenta in the interval $[-\pi, \pi]$.

---

[3]Parameters $h$, $\{\lambda_n^+\}_n$, and $\{\lambda_n^-\}_n$ are the Lagrange multipliers that correspond to constraints of fixed $\mathbf{S}_{\mathrm{st}}^z$, $\{\mathbf{Q}_n^+\}_n$, and $\{\mathbf{Q}_n^-\}_n$, respectively.

**Yang-Yang entropy.** We call Yang-Yang entropy the thermodynamic limit of $\frac{2}{L}\log\Omega$, where $\Omega$ is the size of the space associated with the macrostate. In our specific case, it is given by the sum of Eqs (31) and (32), namely,

$$s_{\text{YY}}[\mu, m_{\text{st}}^z, \rho] = \xi^{-1}\Big[\frac{1-\xi m_{\text{st}}^z}{2}H\big(\tfrac{2\mu}{1-\xi m_{\text{st}}^z}\big) + \frac{1+\xi m_{\text{st}}^z}{2}H\big(\tfrac{2\mu}{1+\xi m_{\text{st}}^z}\big)\Big] + \int_{-\pi}^{\pi}\mathrm{d}p\,\rho_t H\big(\tfrac{\rho(p)}{\rho_t}\big).$$

(33)

Here, $\xi$ and $\rho_t$ must be interpreted as the functionals of $\rho(p)$ and $\mu$ shown in Eqs (25) and (26), respectively.

We can now work out the variation of $\mathfrak{f}_{\text{loc}}[\mu, m_{\text{st}}^z, \rho]$. To that aim, we represent the *local* macrostate as

$$\boldsymbol{\rho} = \frac{e^{h\mathbf{S}_{\text{st}}^z - \mathbf{Q}}}{\text{tr}[e^{h\mathbf{S}_{\text{st}}^z - \mathbf{Q}}]},$$

(34)

where $\mathbf{Q}$ can be any linear combination of the local charges forming the integrable hierarchy described in Section 1.4, namely, $\mathbf{S}^z$ or $\mathbf{Q}_n^{\pm}$. According to Eq. (24), the expectation value of the charge $\mathbf{Q}$ can be written as

$$\frac{2}{L}\text{tr}[\boldsymbol{\rho}\mathbf{Q}] \xrightarrow{\text{TD}} \int_{-\pi}^{\pi}\mathrm{d}p\rho(p)q(p),$$

(35)

where $q(p)$ is its single-particle eigenvalue. By definition, the staggered magnetisation per unit macrosite is equal to $m_{\text{st}}^z$, the generalised free energy thus reads

$$\mathfrak{f}_{\text{loc}}[\mu, m_{\text{st}}^z, \rho] = \int_{-\pi}^{\pi}\mathrm{d}p\rho(p)q(p) - hm_{\text{st}}^z - s_{\text{YY}}[\mu, m_{\text{st}}^z, \rho].$$

(36)

We can readily find the minimum of the functional $\mathfrak{f}_{\text{loc}}[\mu, m_{\text{st}}^z, \rho]$. It is reached in the state with the staggered magnetisation

$$m_{\text{st}}^z = \xi^{-1}\frac{\sqrt{\mu^2 + (1-\mu)^2\sinh^2 h} - \mu\cosh h}{\sinh h} \qquad \big(m_{\text{st}}^z \in [-1, 1]\big),$$

(37)

and the filling function

$$n(p) = \frac{1}{1 + e^{q(p)-w}}, \qquad w = \frac{1}{2}\log\frac{1-(\xi m_{\text{st}}^z)^2}{(1-2\mu)^2 - (\xi m_{\text{st}}^z)^2},$$

(38)

where $\mu$ is implicitly defined as the solution to the equation

$$\int_{-\pi}^{\pi}\frac{\mathrm{d}p}{2\pi}\log\big(1 + e^{-q(p)+w}\big) = \log\frac{4\mu^2}{(1-2\mu)^2 - (\xi m_{\text{st}}^z)^2}.$$

(39)

Alternatively, we can interpret these equations as the statement that a local macrostate with given local integrals of motion satisfies

$$\int_{-\pi}^{\pi}\frac{\mathrm{d}p}{2\pi}\log\big(1 + \tfrac{\mu}{\xi} - 2\pi\rho(p)\big) = -\log\frac{4\mu^2\xi}{\big((1-2\mu)^2 - (\xi m_{\text{st}}^z)^2\big)(\xi + \mu)}.$$

(40)

The latter equation can be used to express $\mu$ as a functional of $\rho(p)$ and $m_{\text{st}}^z$; this is the point of view used when describing the late-time dynamics after quantum quenches. There, the initial state fixes the integrals of motion, i.e., in our specific case, $\rho(p)$ and $m_{\text{st}}^z$.

In Section 5.2 we will use Eq. (39) to show that the locally quasistationary state emerging at late times after joining two thermal states is *not* locally equivalent to a local macrostate. This eventually necessitates the generalisation of local macrostates to quasilocal ones, which we discuss in Section 2.2.2.

**Range of $\mu$ in local macrostates.** The height $\mu[m_{\text{st}}^z, \rho]$ of the surface characterised by Eq. (40) extends over a restricted interval of $\mu$. In particular, we have

$$\frac{4\mu^2\xi}{((1-2\mu)^2-(\xi m_{\text{st}}^z)^2)(\xi+\mu)} = \exp\Big[-\int_{-\pi}^{\pi}\frac{dp}{2\pi}\log\big(1+\tfrac{\mu}{\xi}-2\pi\rho(p)\big)\Big] \geq \frac{\xi+\mu-1}{\xi}, \quad (41)$$

where we used the Jensen's inequality $\int_{-\pi}^{\pi}\frac{dk}{2\pi}f(g(k)) \geq f\big(\int_{-\pi}^{\pi}\frac{dk}{2\pi}g(k)\big)$ for the convex function $f(x) = -\log(1-x)$. We note that the inequality in Eq. (41) is saturated for $\rho(p) = \frac{1}{2\pi\xi}$, which is a physical value for the root density. Thus, for generic $\rho(p)$, the bound on $\mu$ that originates in Eq. (41) cannot be improved.

The parameter $\xi$ is also constrained: we observe $|m_{\text{st}}^z| \leq 1 - |m^z|$, where $m^z = \frac{2}{L}\langle \mathbf{S}^z \rangle = \xi^{-1} - 1$ is the magnetisation per macrosite. The resulting bound reads

$$\xi \geq \frac{1}{2 - m_{\text{st}}^z}. \quad (42)$$

Regions of allowed $\mu$ and $\xi$, for different values of $m_{\text{st}}^z$, are plotted in Fig 1.

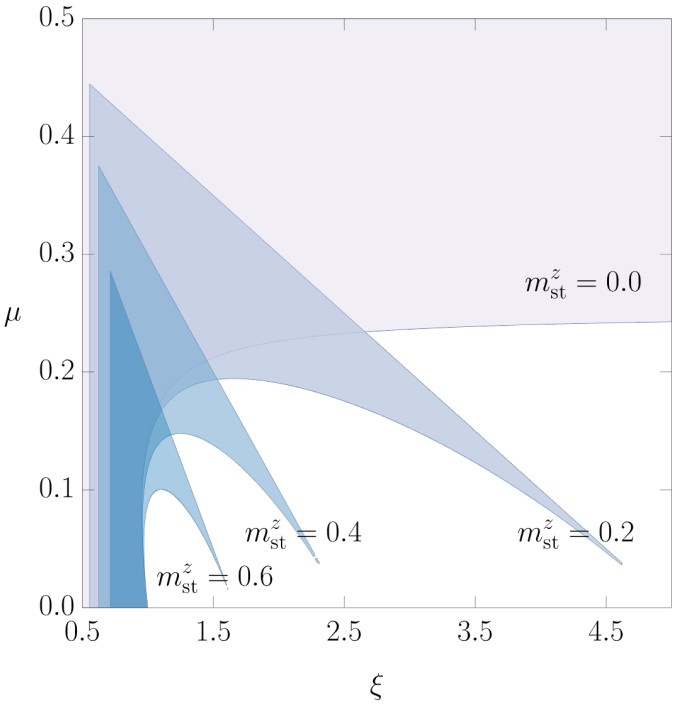

Figure 1: Bounds on $\mu$ and $\xi$, imposed by inequalities (41) and (42), for different values of staggered magnetisation $m_{\text{st}}^z$.

For less generic states it is instead useful to start from Eq. (39) and use the Jensen's inequality for the convex function $f(x) = \log(1 + e^w e^{-x})$. In that case one has

$$\frac{4\mu^2}{(1-2\mu)^2-(\xi m_{\text{st}}^z)^2} = \exp\Big[\int_{-\pi}^{\pi}\frac{dp}{2\pi}\log\big(1+e^w e^{-q(p)}\big)\Big] \geq 1 + e^w e^{-\int_{-\pi}^{\pi}\frac{dp}{2\pi}q(p)}. \quad (43)$$

For $\int_{-\pi}^{\pi}\frac{dp}{2\pi}q(p) = 0$, which corresponds, for example, to thermal states, plugging Eq. (37) into Eq. (43) yields

$$\mu \geq \frac{1}{1 + 2\cosh h}. \quad (44)$$

In the specific case of a one-site shift invariant macrostate, this bound reduces to $\mu \geq 1/3$.

### 2.2.2 A class of quasilocal macrostates

Equation (40) manifests the limitations of considering local macrostates: $\mu = M/N$ is not independent of the other parameters, i.e., it is a functional $\mu[m_{\mathrm{st}}^z, \rho]$. Nevertheless, its value can still be changed by incorporating into the macrostate the quasilocal charge $\mathbf{M}$, reported in Eq. (16). Such a *quasilocal macrostate* is represented by a canonical ensemble of the form

$$\rho = \frac{e^{\chi \mathbf{M} + h \mathbf{S}_{\mathrm{st}}^z - \mathbf{Q}}}{\mathrm{tr}[e^{\chi \mathbf{M} + h \mathbf{S}_{\mathrm{st}}^z - \mathbf{Q}}]}, \tag{45}$$

where $\mathbf{Q}$ is, again, any combination of local charges $\mathbf{Q}_n^{\pm}$ and $\mathbf{S}^z$. The generalised free energy is now given by

$$\mathfrak{f}_{\text{q-loc}}[\mu, m_{\mathrm{st}}^z, \rho] = \int_{-\pi}^{\pi} dp \, \rho(p) q(p) - \chi \mu \int_{-\pi}^{\pi} dp \, \rho(p) - h m_{\mathrm{st}}^z - s_{\mathrm{YY}}[\mu, m_{\mathrm{st}}^z, \rho]. \tag{46}$$

The minimum of the functional $\mathfrak{f}_{\text{q-loc}}[\mu, m_{\mathrm{st}}^z, \rho]$ is the state in which staggered magnetisation and filling function are again given by Eqs (37) and (38), respectively, while $\mu$ is now implicitly defined as the solution to the equation

$$\int_{-\pi}^{\pi} \frac{dp}{2\pi} \log\left(1 + e^{-q(p) + w}\right) = \log \frac{4\mu^2}{(1 - 2\mu)^2 - (\xi m_{\mathrm{st}}^z)^2} - \chi. \tag{47}$$

Contrary to Eq. (39), the additional parameter $\chi$ allows us to vary $\mu$ (almost) independently of the other integrals of motion.

We point out that Eq. (45) does not describe the most general quasilocal macrostate, as there are still infinitely many quasilocal charges in the configuration space [1]. It will nevertheless suffice to describe the late-time behaviour after the junction of two local macrostates (e.g., thermal states). More generally, indicating with $\mathbf{C}$ a generic quasilocal charge in the configuration space, the most general quasilocal macrostate can be represented as follows:

$$\rho = \frac{e^{\chi \mathbf{M} + \mathbf{C} - \mathbf{Q}}}{\mathrm{tr}[e^{\chi \mathbf{M} + \mathbf{C} - \mathbf{Q}}]}. \tag{48}$$

Like the staggered spin along the $z$-axis, the charge $\mathbf{C}$ affects the entropy in the configuration space, which, for given $\mathbf{C}$, becomes a functional of $\mu$, $\xi$, and $\frac{2}{L}\langle \mathbf{C} \rangle$. This, in turn, moves the minimum of the generalised free energy without however affecting the general functional form of $n(p)$, given on the left-hand side of Eq. (38).[4]

### 2.3 Thermal states

The Gibbs canonical ensemble at temperature $T = 1/\beta$ is described by the density matrix $\rho = e^{-\beta \mathbf{H}}/\mathrm{tr}[e^{-\beta \mathbf{H}}]$, and it corresponds to setting $h = 0$ and $q(p) = \beta E(p)$ in Eq. (34). Here $E(p) = 4J \cos p$ denotes the single-particle eigenvalue of the energy. The TBA equations can be solved numerically to obtain, for example, the macrosite energy density as a function of the inverse temperature $\beta$. In the latter case, comparison with the DMRG-based numerical simulation confirms the TBA prediction, as evident in Fig. 2.

Consistently with the results of Ref. [1], we find that $\mu$ tends to $1/2$ as the temperature drops to zero. In addition, Fig. 3 clearly shows that the limit is approached exponentially fast in $\beta$. It is then convenient to parametrise $\mu$ as

$$\mu = \frac{1}{2}(1 - e^{-4J\beta \cos k(\beta)}). \tag{49}$$

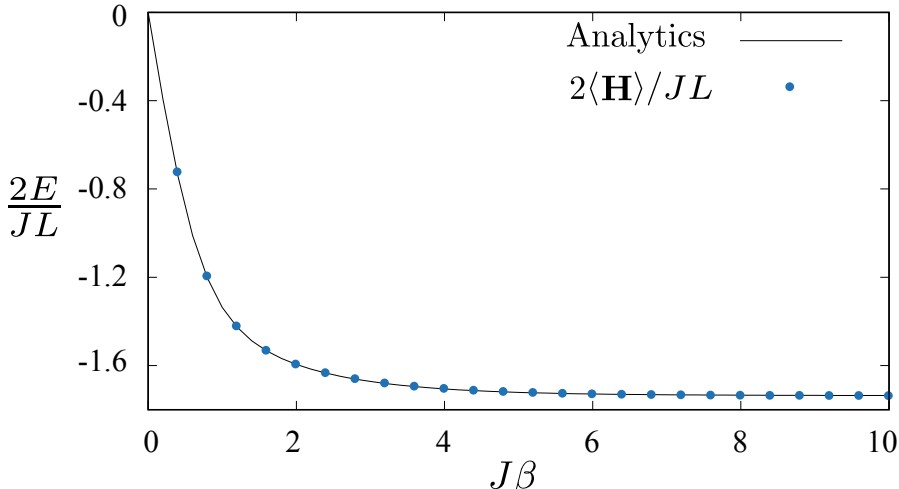

Figure 2: Energy per macrosite as a function of the inverse temperature $\beta$. The results of the ancilla-based DMRG simulation reproduce those of the thermodynamic Bethe Ansatz calculation up to $10^{-4}$. See Appendix B, in particular Fig. 11, for more details on numerical simulations.

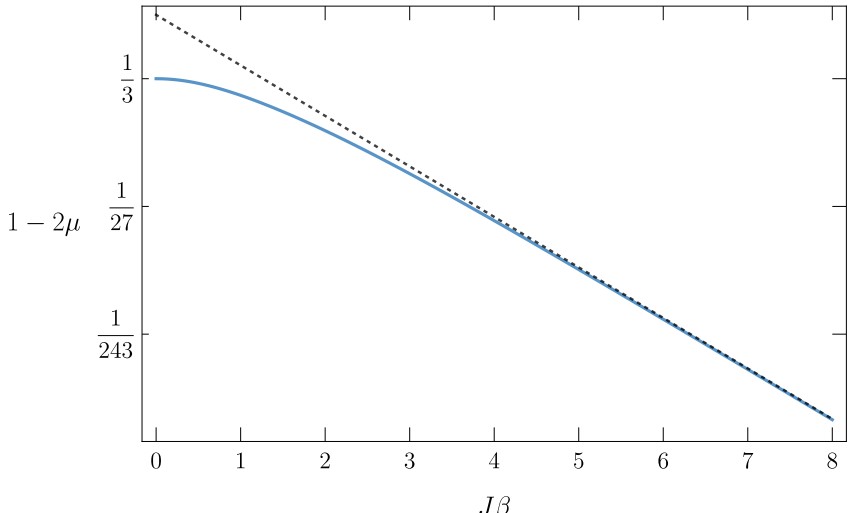

Figure 3: Parameter $\mu$ as a function of $\beta$ (blue curve). The dotted line captures the asymptotic behaviour at low temperature.

Plugging this Ansatz into Eq. (39), splitting the integration domain, and changing the integration variables into linear functions of $\cos p$ results in the identity

$$\int_0^{1-\cos k(\beta)} \frac{\log(1+e^{-4J\beta x})\mathrm{d}x}{\pi\sqrt{1-[x+\cos k(\beta)]^2}} + \int_0^{1+\cos k(\beta)} \frac{\log(1+e^{-4J\beta x})\mathrm{d}x}{\pi\sqrt{1-[x-\cos k(\beta)]^2}} +$$

$$+ \frac{4J\beta}{\pi}\int_0^{1+\cos k(\beta)} \frac{x\mathrm{d}x}{\sqrt{1-[x-\cos k(\beta)]^2}} = 8\beta J\cos k(\beta) + 2\log(1-e^{-4J\beta\cos k(\beta)}). \quad (50)$$

---

[4]Note, however, that the parameter $w$ on the right-hand side of Eq. (38) can still be affected. For instance, in the local macrostate with a fixed staggered magnetisation it depends on $m_{\text{st}}^z$.

In Appendix A it is shown that the first two terms on the left-hand side of Eq. (50) asymptotically behave as

$$\int_0^{1-z} \frac{dx \log(1 + e^{-4J\beta x})}{\pi\sqrt{1 - [x+z]^2}} = \sum_{n=0}^{j} \frac{\arcsin^{(n+1)}(z)(1 - 2^{-n-1})\zeta(n+2)}{\pi(4J\beta)^{n+1}} + \mathcal{O}((J\beta)^{-j-2}), \quad (51)$$

where $\zeta$ is the Riemann zeta function and we performed a Sommerfeld expansion of the integrand. On the other hand, we can analytically integrate the third term on the left-hand side of Eq. (50) to obtain the asymptotic expansion

$$\pi + k(\beta) - \tan k(\beta) = \sum_{n=0}^{j-1} \frac{(2 - 2^{-2n})\zeta(2n+2)}{(4J\beta)^{2n+2}} \frac{\arcsin^{(2n+1)}[\cos k(\beta)]}{\cos k(\beta)} + \mathcal{O}((J\beta)^{-2j-2}),$$

$$(52)$$

which is effective as long as $j \ll J\beta / \log(J\beta)$.

**Ground state.**  In the limit $T \to 0$ we can approximate $n(p) = [1 + e^{4J\beta(\cos p - \cos k(\beta))}]^{-1}$ by a characteristic function

$$n(p) \xrightarrow{\beta \to \infty} \chi_{[-\pi, -k_F) \cup [k_F, \pi)} = \begin{cases} 1, & \text{if } k \in [-\pi, -k_F) \cup [k_F, \pi), \\ 0, & \text{if } k \in [-k_F, k_F), \end{cases} \quad (53)$$

where $k_F \equiv \lim_{\beta \to \infty} k(\beta)$ is the Fermi momentum. In addition, in this limit Eq. (52) reduces to a simple transcendental equation

$$k_F = \arctan(k_F + \pi). \quad (54)$$

The solution to Eq. (54) exists and is such that $\cos k_F$ is nonzero: consistently with the numerical observation and with Ansatz (49), we find that the ground state has $\mu = 1/2$. The energy per unit macrosite is now computed by enforcing the replacement (53) in the root density $\rho(p) = \rho_t n(p)$, where $\rho_t$ is reported in Eq. (27). We finally obtain

$$\frac{2E_{GS}}{L} \xrightarrow{TD} 4J \int_{-\pi}^{\pi} dk \cos k \lim_{\beta \to \infty} \rho(k) = -\frac{8J \sin k_F}{k_F + \pi} = -8J \cos k_F, \quad (55)$$

where Eq. (54) has been used in the last equality. The numeric value of the ground state energy per unit macrosite reads $-8J \cos k_F \approx -1.73787J$ and perfectly agrees both with numerical investigations and with the thermodynamic limit of the analytic result that was obtained in our first work [1] focused on finite chains. We can therefore conclude that the thermodynamic ground state matches the actual ground state. Finally, we report the number of particles per unit macrosite, $\xi^{-1}$, which is given by $2N_{GS}/L \xrightarrow{TD} 2(\pi - k_F)/(\pi + k_F) \approx 0.796625$.

**Low temperature.**  For large but finite $\beta$ the right-hand side of Eq. (52) can not be neglected. Instead, the leading finite-temperature corrections are obtained by setting $j = 2$ in the latter equation, yielding

$$\pi + k(\beta) - \tan k(\beta) = \frac{\pi^2}{3 \sin(2k_F)} \frac{1}{(4J\beta)^2} + \frac{7\pi^4}{360} \frac{1 + 2\cos^2 k_F}{\sin^5 k_F \cos k_F} \frac{1}{(4J\beta)^4} + \mathcal{O}((J\beta)^{-6}). \quad (56)$$

By expanding $k(\beta) = k_\mathrm{F} + \sum_{n>1} c_n (4\beta J)^{-n}$, for some real coefficients $c_n$, we can then obtain $\cos k(\beta)$ as a function of Fermi momentum $k_\mathrm{F}$:

$$\cos k(\beta) = \cos k_\mathrm{F} + \frac{\pi^2 \cos k_\mathrm{F}}{6 \sin^2 k_\mathrm{F}} \frac{1}{(4J\beta)^2} + \frac{\pi^4 \cos k_\mathrm{F} (43 + 9 \cos(2k_\mathrm{F}))}{720 \sin^6 k_\mathrm{F}} \frac{1}{(4J\beta)^4} + \mathcal{O}((J\beta)^{-6}). \quad (57)$$

Using this in Eq. (49) we see that $\mu$ remains exponentially close to $1/2$.

As shown in Appendix A, $\xi(\beta)$ and the energy $E(\beta)$ per unit macrosite can be obtained by carrying out expansions analogous to the one in Eq. (51). In particular we have

$$m^z(\beta) = \xi^{-1}(\beta) - 1 = \frac{\pi - 3k_\mathrm{F}}{\pi + k_\mathrm{F}} + \frac{4\pi^3 \cos^3 k_\mathrm{F}}{3 \sin^5 k_\mathrm{F} (4J\beta)^2} + \frac{\pi^5 \cos^3 k_\mathrm{F} (74 + 29 \cos(2k_\mathrm{F}))}{45 \sin^9 k_\mathrm{F} (4J\beta)^4} + \mathcal{O}((4J\beta)^{-6}), \quad (58)$$

and

$$E(\beta) = -8J \cos k_\mathrm{F} + \frac{4J\pi^2 \cos k_\mathrm{F}}{3 \sin^2 k_\mathrm{F} (4J\beta)^2} + \frac{\pi^4 J \cos k_\mathrm{F} (43 + 29 \cos(2k_\mathrm{F}))}{30 \sin^6 k_\mathrm{F} (4J\beta)^4} + \mathcal{O}((J\beta)^{-6}), \quad (59)$$

the first terms reproducing the thermodynamic limits of the ground state values $\frac{2N_\mathrm{GS}}{L} - 1$ and $2E_\mathrm{GS}/L$, respectively.

We also report the first orders of the asymptotic low-temperature expansion of the specific heat,

$$c_V(\beta) \equiv -\beta^2 \partial_\beta E(\beta) = \frac{2\pi^2 \cos k_\mathrm{F}}{3 \sin^2 k_\mathrm{F} (4J\beta)} + \frac{\pi^4 \cos k_\mathrm{F} (43 + 29 \cos(2k_\mathrm{F}))}{30 \sin^6 k_\mathrm{F} (4J\beta)^3} + \mathcal{O}((J\beta)^{-5}), \quad (60)$$

which is shown in Fig. 4. We point out that the sub-leading correction, i.e., $\mathcal{O}((J\beta)^{-4})$ in the energy and in the magnetisation, is practically irrelevant, somehow manifesting the asymptotic character of the expansion.

**High temperature.** In the limit of infinite temperature $\beta = 0$ one immediately obtains $\mu = 1/3$, $n(p) = 3/4$, and hence $\rho(p) = \frac{1}{2\pi}$. This is consistent with all traceless local operators having zero expectation value in the infinite-temperature state. An example is given by the conserved charges with single-particle expectation values given in Eq. (18).

At small, but finite $\beta$, $\mu$ remains close to $1/3$, and we are about to show that the deviation scales as $\beta^2$. It is then convenient to parametrise $\mu$ as

$$\mu = \frac{1}{3} + \frac{J^2 \beta^2}{18} z(\beta), \quad (61)$$

where $z(\beta)$ is to be computed. Plugging this into Eq. (39) yields

$$\int_{-\pi}^{\pi} \frac{\mathrm{d}p}{2\pi} \log\left(1 - \frac{J^2 \beta^2}{12} z(\beta) + \frac{3}{4}(e^{-4J\beta \cos(p)} - 1)\right) = \log \frac{\left(1 + \frac{J^2 \beta^2}{6} z(\beta)\right)^2}{1 - \frac{J^2 \beta^2}{3} z(\beta)}, \quad (62)$$

which can easily be expanded about $J\beta = 0$. In particular we have

$$z(\beta) = 1 - \frac{1}{3}(J\beta)^2 + \mathcal{O}((J\beta)^4), \quad (63)$$

confirming the quadratic scaling of the deviation of $\mu$ from the infinite-temperature value $1/3$.

Since, in the limit, the root density remains smooth, we can directly expand it for small $J\beta$, and find

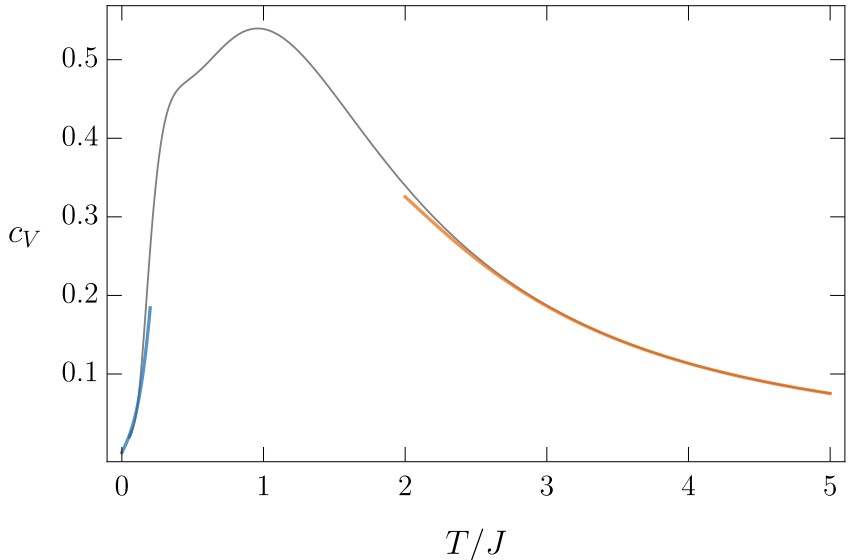

Figure 4: The specific heat as a function of the temperature (black solid line). The solid blue and orange lines represent the first orders of the low-temperature (given in Eq. (60)) and the high-temperature (given in Eq. (65)) asymptotic expansions, respectively.

$$\rho(p) = \frac{1}{2\pi}\Big[1 - J\beta\cos p - J^2\beta^2\cos^2 p + \frac{J^3\beta^3}{12}(6\cos p + \cos(3p)) + J^4\beta^4\Big(\frac{1}{8} + \frac{5\cos^4 p}{3}\Big) +$$
$$+ \frac{J^5\beta^5}{240}\big(-37 + 34\cos(2p) + 26\cos(4p)\big)\Big] + O((J\beta)^6). \quad (64)$$

The asymptotic values of the energy per unit macrosite and the specific heat now readily follow (see Fig. 4):

$$E(\beta) = -2J^2\beta + J^4\beta^3 - \frac{J^6\beta^5}{6} + O((J\beta)^7),$$
$$c_V(\beta) = 2(J\beta)^2 - 3(J\beta)^4 + \frac{5}{6}(J\beta)^6 + \mathcal{O}((J\beta)^8). \quad (65)$$

Analogously, the magnetisation per macrosite is given by

$$m^z = \xi^{-1} - 1 = -\frac{1}{2}(J\beta)^2 + \frac{3}{4}(J\beta)^4 + O((J\beta)^6). \quad (66)$$

## 3  Elementary excitations

We define here the elementary excitations of the model. In Ref. [1] we have classified the Bethe states with a set of $N$ momenta that depend on the particle configuration $B_N = \{(b_1, \ldots, b_N)\}_c$ only through $N$ and $M$. We distinguish two types of elementary excitations:

A: creation or annihilation of a momentum along with a Bethe shift of the remaining momenta;

B: global "fractional" shift of the momenta.

A close look at the solution (12) to the Bethe equations reveals the following. Excitations of type A are obtained by adding or removing a quantum number $I_\ell$, and choosing $I_0$ so as to make the change in $\varphi$ negligible in the thermodynamic limit. These are the standard excitations, analogous to the ones in the XXZ model, which are defined through addition or removal of a Bethe-Takahashi quantum number. On the other hand, the excitation of type B can be interpreted as a global shift of quantum numbers $I_\ell$ by a fractional amount. It is achieved by changing $\varphi$ (through $I_0$).

|  | $B_N$ | $I_0$ | $\{I_1,\ldots,I_N\}$ |
|---|---|---|---|
| eigenstate | $\{(\ldots,1,1,0,1,0,\ldots)\}_c$ | 0 | $\{1,3,4,5,6,\ldots\}$ |
| hole excitation (type A) | $\{(\ldots,1,1,\cancel{0},1,0,\ldots)\}_c$ | 1 | $\{1,3,4,\cancel{5},6,\ldots\}$ |
| particle excitation (type A) | $\{(\ldots,1,1,0,\mathbf{1},1,0,\ldots)\}_c$ | 3 | $\{1,\mathbf{2},3,4,5,6,\ldots\}$ |
| excitation of type B | $\{(\ldots,1,1,0,1,0,\ldots)\}_c$ | $\mathbf{N/3}$ | $\{1,3,4,5,6,\ldots\}$ |

Figure 5: Examples of elementary excitations over an eigenstate assuming $g = 1$ both before and after the excitation. In the example above the hole excitation decreases $M$ by 1, whereas the particle excitation does not change $M$.

**Excitations of type A.** In terms of the rational quantum numbers, $\{J_1,\ldots,J_N\}$, excitations of type A correspond to removing or adding a quantum number at an integer distance from the others;

$$
\begin{aligned}
&\text{hole excitation: } \{J_1,\ldots,J_N\} \to \{J_1,\ldots,J_{\ell-1},J_{\ell+1},\ldots J_N\} + \mathcal{O}(N^{-1}), \\
&\text{particle excitation: } \{J_1,\ldots,J_N\} \to \{J_1,\ldots,J_{\ell-1},J',J_\ell,\ldots J_N\} + \mathcal{O}(N^{-1}), \ J'-J_j \in \mathbb{Z}.
\end{aligned}
\tag{67}
$$

A hole excitation corresponds to removing an integer $I_\ell$ or, equivalently, the corresponding rational quantum number $J_\ell$. Since $N \to N-1$, this is necessarily accompanied by a change in the configuration; we propose to update it by removing an elementary particle – see, e.g., Figure 5. As a consequence, $M$ can either remain unchanged or decrease by 1, i.e. $\Delta M \in \{0,-1\}$. More quantitatively, representing this operation by an operator $\mathbf{C}(p_\ell; b_j)$ acting on the eigenstate, we have

$$
\begin{aligned}
&\mathbf{C}(p_\ell; b_j) \,|\{p_1,\ldots,p_N\}; \{(b_1,\ldots,b_N)\}_c\rangle \propto \\
&|\{p_1 + \delta p_1,\ldots,p_{\ell-1} + \delta p_{\ell-1}, p_{\ell+1} + \delta p_{\ell+1},\ldots,p_N + \delta p_N\}; \{(b_1,\ldots,b_{j-1},b_{j+1},\ldots,b_N)\}_c\rangle.
\end{aligned}
\tag{68}
$$

Here, using Bethe equations (12), we find

$$
\frac{L}{2}\delta\vec{p} = W\vec{p} - \mu p_\ell \vec{u} + \mathcal{O}(N^{-1}),
\tag{69}
$$

where $\vec{u} = [1]_{j=1}^{N-1}$, $\vec{p} = [p_1,\ldots,p_{\ell-1},p_{\ell+1},\ldots,p_N]$, $\delta\vec{p} = [\delta p_1,\ldots,\delta p_{\ell-1},\delta p_{\ell+1},\ldots,\delta p_N]$, and we have defined

$$
W = \frac{2\xi}{L}\Big(\mu - \frac{|\Delta M|}{1 + \frac{\mu}{\xi}}\Big)\vec{u} \otimes \vec{u} + \frac{|\Delta M|}{1 + \frac{\mu}{\xi}}\mathbb{1}.
\tag{70}
$$

Analogously, a particle excitation corresponds to adding an integer $I_\ell$. The configuration is updated by adding an elementary particle. As a consequence, $N \to N+1$ and $M$ can either remain unchanged or increase by 1, i.e., $\Delta M \in \{0,1\}$. Representing this operation by an operator $\mathbf{B}(p_\ell; b_j)$ acting on the eigenstate, we have

$$\mathbf{B}(p_\ell; b_j) | \{p_1, \ldots, p_{\ell-1}, p_{\ell+1}, \ldots, p_N\}; \{(b_1, \ldots, b_{j-1}, b_{j+1}, \ldots, b_N)\}_c\rangle \propto$$

$$| \{p_1 - \delta p_1, \ldots, p_{\ell-1} - \delta p_{\ell-1}, p_\ell, p_{\ell+1} - \delta p_{\ell+1}, \ldots, p_N - \delta p_N\}; \{(b_1, \ldots, b_N)\}_c\rangle , \quad (71)$$

where $\delta\vec{p}$ is still given by Eq. (69).

Almost always, both before and after the excitation, $g = 1$, and thus $I_0$ can be left unchanged. If, before or after the excitation, $g \neq 1$, but still $g \ll N$, the change in $\varphi$ can always be compensated up to $\mathcal{O}(N^{-1})$ by a change in $I_0$. If, instead, after the excitation $g \propto N$, then the compensation is not always possible and generally such excitation is composite, consisting of an elementary excitation of type B, followed by an elementary excitation of type A.

**Excitation of type B.** An excitation of type B does not modify the configuration and corresponds to an extensive change of $I_0$ – see Figure 5. In terms of the rational quantum numbers $\{J_1, \ldots, J_N\}$ we have

$$\{J_1, \ldots, J_N\} \rightarrow \{J_1 + \tfrac{\delta\varphi}{2\pi}, \ldots, J_N + \tfrac{\delta\varphi}{2\pi}\}. \quad (72)$$

If we agree on representing this operation by an operator $e^{i\frac{2\varphi}{L}\mathbf{X}}$ acting on the eigenstate, we have[5]

$$e^{i\frac{2\delta\varphi}{L}\mathbf{X}} | \{p_1, \ldots, p_N\}; \{(b_1, \ldots, b_N)\}_c\rangle \propto | \{p_1 + \tfrac{2\delta\varphi}{L}, \ldots, p_N + \tfrac{2\delta\varphi}{L}\}; \{(b_1, \ldots, b_N)\}_c\rangle . \quad (73)$$

We note that $e^{i\frac{4\pi}{L}\mathbf{X}}$ can be written in terms of excitations of type A – *cf.* Eqs (67) and (72):

$$e^{i\frac{4\pi}{L}\mathbf{X}} | \{p_1, \ldots, p_N\}; \{(b_1, \ldots, b_N)\}_c\rangle \propto$$

$$\prod_{\ell=1}^{N} \mathbf{B}(p_\ell + \tfrac{4\pi}{L}; b_{j_\ell})\mathbf{C}(p_\ell; b_{j_\ell}) | \{p_1, \ldots, p_N\}; \{(b_1, \ldots, b_N)\}_c\rangle . \quad (74)$$

We warn the reader that we conceived these elementary excitations so as to form a basis of the space of excitations and to preserve the macroscopic properties of the excited states. We did not investigate, however, whether the matrix elements of local observables could be nonzero even between states differing in a macroscopically large number of elementary excitations.

**Dressed charges**

The changes in the expectation values of the local charges (17) after an elementary excitation are of the order $\mathcal{O}(1)$. It is customary to interpret such discrepancies as the charges carried by the excitation. In interacting systems these values depend on the state and are usually called "dressed charges". For example, after a hole excitation of type A, generated by operator $\mathbf{C}(p_\ell; b_j)$, the integral of motion with a single-particle eigenvalue $q(p)$ changes as

$$-q^{\mathrm{dr}}(p_\ell; b_j) = \Delta Q = \sum_{\substack{j=1 \\ j\neq\ell}}^{N} q(p_j + \delta p_j) - \sum_{j=1}^{N} q(p_j) = -q(p_\ell) + \vec{q}' \cdot \delta\vec{p} + \mathcal{O}(N^{-1}), \quad (75)$$

and hence, using Eq. (69), we find

$$q^{\mathrm{dr}}(k; b_j) \xrightarrow{\mathrm{TD}} q(k) + \mu \frac{k\langle 1\rangle - \langle p\rangle}{\langle 1\rangle} \langle q'(p)\rangle - \frac{|\Delta M_j|}{1 + \mu\langle 1\rangle} \frac{\langle q'(p)p\rangle \langle 1\rangle - \langle q'(p)\rangle \langle p\rangle}{\langle 1\rangle} . \quad (76)$$

Here, we have extended the notation $\langle\bullet\rangle$, for the average of an operator in the macrostate described by $\rho(p)$, to functions of momenta:

$$\langle f(p)\rangle = \int_{-\pi}^{\pi} \mathrm{d}p\, \rho(p) f(p).[6] \quad (77)$$

---

[5]Notation follows from the standard way of defining the momentum-shift operator.
[6]When unambiguous, we will use simply $\langle f\rangle = \langle f(p)\rangle$, in order to ease the notation.

The dressed momentum $\wp$ thus reads

$$\wp(k; b_j) \equiv \mathrm{id}^{\mathrm{dr}}(k) = \left(1 + \frac{\mu}{\xi}\right)k - \mu \int_{-\pi}^{\pi} \mathrm{d}p \rho(p) p \, . \qquad (78)$$

Comparing this with Eq. (26), we obtain the standard relation between the dressed momentum derivative and the total root density, i.e., $2\pi\rho_{\mathrm{t}} = \partial_k \wp(k; b_j)$.

Analogously, the dressed energy $\varepsilon \equiv E^{\mathrm{dr}}$ of the excitation corresponds to $q(p) = 4J \cos p$ and is given by

$$\varepsilon(k; b_j) = 4J\Big[\cos k - \mu \frac{k \langle 1 \rangle - \langle p \rangle}{\langle 1 \rangle} \langle \sin p \rangle + \frac{|\Delta M_j|}{1 + \mu \langle 1 \rangle} \frac{\langle p \sin p \rangle \langle 1 \rangle - \langle \sin p \rangle \langle p \rangle}{\langle 1 \rangle}\Big] \, . \qquad (79)$$

In the specific case of a thermal state at temperature $\beta^{-1}$ the expectation value of any charge that is odd under spatial reflections zeroes, so we have

$$\varepsilon_\beta(k; b_j) = 4J \cos k + 4J \frac{|\Delta M_j|}{1 + \frac{\mu}{\xi}} \langle p \sin p \rangle_\beta \, , \qquad (80)$$

where $\langle \bullet \rangle_\beta$ signifies that the average is taken in a thermal macrostate. Contrary to the XXZ model, the dressed energy of an elementary excitation does not match $\beta^{-1} \log\left(\frac{1}{n(p)} - 1\right)$, which is unhappily called "dressed energy" as well[7]. In a thermal state, however, their derivatives with respect to the momentum coincide.

For the sake of completeness, we finally report the derivative of the dressed charge with respect to the momentum (rapidity), which plays a key role in the thermodynamic Bethe Ansatz. It reads

$$\partial_k q^{\mathrm{dr}}(k; b_j) = q'(k) + \mu \int_{-\pi}^{\pi} \mathrm{d}p \rho(p) q'(p) \, , \qquad (81)$$

where $(\bullet)'$ denotes the derivative.

## 4 Hydrodynamics with a minimal set of charges

This section develops the tools necessary to describe time evolution of inhomogeneous states in the folded XXZ model. A theory, now known as "generalised hydrodynamics", was proposed in the seminal papers [5, 6] to describe the late-time behaviour of the expectation values of local observables in integrable systems that evolve in time in the presence of inhomogeneities. Specifically, it was assumed that one can describe the time evolution of a large class of inhomogeneous states by lifting the root densities that characterise stationary states into functions of space and time (in addition to the momentum). In this way, the concept of local equilibrium is generalised to integrable systems, where the stationary states are characterised by infinitely many conservation laws.

The structure underlying local equilibrium in noninteracting spin chains was clarified in Ref. [25]. There, it was shown that generalised hydrodynamics can be interpreted as a phase-space description of the dynamics in a subspace of the Hilbert space that is invariant under time evolution. When scrutinised locally, the states belonging to that subspace are quasistationary –

---

[7] By identifying the filling function with a Fermi-Dirac distribution, $\beta^{-1} \log\left(\frac{1}{n(p)} - 1\right)$ represents the single-particle energy.

for that reason they have been called "locally quasistationary states" [26] – and the generalised hydrodynamic equation of motion is simply the projection of the Schrödinger equation onto the invariant subspace.

In interacting integrable systems the structure is less transparent and the aforementioned invariant subspaces have not yet been identified. The theory is therefore still based on the weaker assumption that, after a long-enough time, the unspecified degrees of freedom that can not be described by space-time dependent root densities become irrelevant. This corresponds to identifying the leading order(s) of an asymptotic expansion in the degree of the inhomogeneity. To that aim, we first need to define the charge densities and the currents of the integrable hierarchy introduced in Section 1.4. We will then develop first-order generalised hydrodynamics in the dual folded XXZ model.

## 4.1 Charge densities

A local charge $\mathbf{Q}$ that commutes with the shift operator $e^{i\mathbf{P}}$, where $\mathbf{P}$ is the operator of the total momentum, can always be represented as a sum of localised operators that differ only in their position: $\mathbf{Q} = \sum_\ell \mathbf{q}_\ell$, with $\mathbf{q}_{\ell+1} = e^{i\mathbf{P}}\mathbf{q}_\ell e^{-i\mathbf{P}}$. The operators $\mathbf{q}_\ell$ are referred to as charge densities, but their definition is not at all unique. For example, if $\mathbf{q}_\ell$ is a charge density, the operator $\mathbf{q}_\ell + \mathbf{z}_\ell - \mathbf{z}_{\ell-1}$ is as well, whatever $\mathbf{z}_\ell$ is. Such an ambiguity can be used to simplify either the definition of the operator or its dynamical equations, the latter point of view being taken, for example, in Ref. [25]. In both cases, it is usually convenient to choose a density with as many symmetries as possible. Simple symmetries that can be easily enforced are generated by operators with equally spaced eigenvalues. Consider, for example, the total magnetisation $\mathbf{S}^z$. The eigenvalues of $\frac{L}{2} + \mathbf{S}^z$ are integers, thus $e^{i\varphi(\frac{L}{2}+\mathbf{S}^z)}$ is $2\pi$-periodic in $\varphi$. This implies that the averaged density

$$\bar{\mathbf{q}} = \int_0^{2\pi} \frac{\mathrm{d}\varphi}{2\pi} e^{i\varphi\mathbf{S}^z} \mathbf{q} e^{-i\varphi\mathbf{S}^z} , \tag{82}$$

commutes with $\mathbf{S}^{z}$[8]. Since a rotation does not change the range of $\mathbf{q}$, the local density $\bar{\mathbf{q}}$ has the same locality properties as $\mathbf{q}$, but is, in addition, $U(1)$-invariant.

As a matter of fact, a similar conclusion can be drawn even if we replace $\mathbf{S}^z$ with a less trivial charge with equally-spaced spectrum, say $\mathbf{A}$. The averaged charge still commutes with $\mathbf{A}$, but its range might increase. Nevertheless, the range can not become arbitrarily large, provided that the local terms of $\mathbf{A}$ decay exponentially with their range. This follows from a result of Ref. [27], which showed that, if $\mathbf{q}$ is a local operator with range $r$, the operator $e^{i\varphi\mathbf{A}}\mathbf{q}e^{-i\varphi\mathbf{A}}$ can be approximated, with exponential accuracy, by an operator with range $r + 2v_{\mathrm{LR}}T$, where $v_{\mathrm{LR}}$ is the Lieb-Robinson velocity of $\mathbf{A}$. Specifically, the error of this approximation, i.e., the part of operator $e^{i\varphi\mathbf{A}}\mathbf{q}e^{-i\varphi\mathbf{A}}$ with a range larger than $r + 2v_{\mathrm{LR}}T$, is exponentially small in $v_{\mathrm{LR}}(T - \varphi)$. This bound can be readily adapted to the averaged charge density by replacing $\varphi$ with its maximum, namely $2\pi$.

In our specific case, the invariance of the configuration can be expressed as the conservation of infinitely many quasilocal charges with equally spaced spectrum (including $\mathbf{S}^z$, $\mathbf{S}_{\mathrm{st}}^z$, and $\mathbf{M}$). It is therefore natural to define charge densities that preserve the configuration. Remarkably, the local densities (19) are already of such form. This holds even if one redefines them by adding a local operator $\mathbf{z}_{n,\ell}^\pm$ that preserves the configuration and satisfies $\sum_{\ell=1}^L \mathbf{z}_{n,\ell}^\pm = 0$.

---

[8]

$$[\bar{\mathbf{q}}, \mathbf{S}^z] = \int_0^{2\pi} \frac{\mathrm{d}\varphi}{2\pi} e^{i\varphi\mathbf{S}^z}[\mathbf{q}, \mathbf{S}^z]e^{-i\varphi\mathbf{S}^z} = \int_0^{2\pi} \frac{\mathrm{d}\varphi}{2\pi} i\partial_\varphi \left(e^{i\varphi\mathbf{S}^z}\mathbf{q}e^{-i\varphi\mathbf{S}^z}\right) = 0 .$$

Concerning the diagonal charges (the ones commuting with any $\boldsymbol{\sigma}_\ell^z$), a sensible choice is to enforce their densities to be diagonal as well. In particular, we will use $\mathbf{s}_\ell^z = \boldsymbol{\sigma}_\ell^z/2$ and

$$\mathbf{m}_\ell = \frac{\boldsymbol{\sigma}_\ell^z}{2} + \frac{1}{2}\prod_{j=1}^{L}\frac{1-\boldsymbol{\sigma}_j^z}{2} + \sum_{n=1}^{L-2}\prod_{j=\ell}^{\ell+n}\frac{\boldsymbol{\sigma}_j^z-1}{2}\,, \tag{83}$$

as diagonal local densities of $\mathbf{S}^z$ and $\mathbf{M}$, respectively. Finally, we mention that the expectation value of $\mathbf{m}_\ell$ can be rewritten in terms of the emptiness formation probability $P_\ell(n) = \langle \prod_{j=\ell}^{\ell+n-1}\frac{1-\boldsymbol{\sigma}_j^z}{2} \rangle$ as follows:

$$\langle \mathbf{m}_\ell \rangle = \frac{1}{2} + \frac{1}{2}P_\ell(L) + \sum_{n=1}^{L-1}(-1)^n P_\ell(n)\,. \tag{84}$$

## 4.2  Currents

A fundamental role in generalised hydrodynamics is played by the continuity equations that relate the local charge densities to the corresponding currents. In a spin chain, they are obtained by applying the Heisenberg equation to the charge restricted to a half-open interval $\Delta X = a[n_-, n_+)$, where $n_-$ and $n_+$ are the integers denoting the boundary sites, while $a$ is the lattice spacing. In particular, the equation reads $\partial_t \sum_{\ell \in \Delta X/a} \mathbf{q}_\ell(t) = i[\mathbf{H}, \sum_{\ell \in \Delta X/a} \mathbf{q}_\ell(t)]$; since the commutator acts nontrivially only around the boundaries of the region, it can also be written as

$$\partial_t \sum_{\ell \in [n_-, n_+)} \mathbf{q}_\ell(t) + \frac{\mathbf{j}_{n_+}(t) - \mathbf{j}_{n_-}(t)}{a} = 0\,. \tag{85}$$

If the region $\Delta X$ includes a mesoscopically large number of sites $((x_+ - x_-)/a \gg 1)$ and the local properties of the state vary in a sufficiently smooth way on a larger scale, the continuity equation can also be expressed in a differential form. Here, the expectation values are interpolated by smooth functions $\langle \mathbf{q}_\ell \rangle(t) = a\langle \mathbf{q} \rangle(x, t)$ and $\langle \mathbf{j}_\ell \rangle(t) = a\langle \mathbf{j} \rangle(x, t)$, with $x = a\ell$, and, according to the trapezoidal rule and the definition of the derivative, we have

$$\frac{\sum_{\ell \in [n_-, n_+)} \langle \mathbf{q}_\ell(t) \rangle}{x_+ - x_-} \sim \int_{x_-}^{x_+} \frac{\mathrm{d}x}{x_+ - x_-} \langle \mathbf{q} \rangle(x, t) - a\frac{\langle \mathbf{q} \rangle(x_+, t) - \langle \mathbf{q} \rangle(x_-, t)}{2(x_+ - x_-)} \xrightarrow{x_+ \to x_-} \langle \mathbf{q} \rangle(x_-, t)\,,$$
$$\frac{\langle \mathbf{j}_{n_+}(t) \rangle - \langle \mathbf{j}_{n_-}(t) \rangle}{a(x_+ - x_-)} = \frac{\langle \mathbf{j} \rangle(x_+, t) - \langle \mathbf{j} \rangle(x_-, t)}{x_+ - x_-} \xrightarrow{x_+ \to x_-} \partial_x \langle \mathbf{j} \rangle(x, t)\Big|_{x=x_-}\,. \tag{86}$$

Importantly, if the ranges of $\mathbf{q}_\ell$ and $\mathbf{j}_\ell$ are both small with respect to $\Delta X/a$, one can ignore the inhomogeneity of the state in the region where $\mathbf{q}_\ell$ and $\mathbf{j}_\ell$ act nontrivially, so that the state can be effectively replaced by a homogeneous one.

In our specific case, there exist local and quasilocal conservation laws that break one-site shift invariance, therefore we can not generally expect the smoothness hypothesis to hold on a mesoscopic scale if $\ell$ is associated with a chain site. The natural choice for the constituents of the unit cell are instead macrosites – the index $\ell$ in Eqs (85) and (86) should then refer to a pair of neighbouring spins. Technically speaking, this is a consequence of the fact that the momentum entering the Bethe equations (21) generates two-site translations. For the sake of clarity, let us introduce the notations $\overline{\mathbf{q}}_{\ell'} = \mathbf{q}_{2\ell'-1} + \mathbf{q}_{2\ell'}$ for the macrosite charge density and

$\overrightarrow{J}_{\ell'}$ for the macrosite current. For example, using definitions (19), the energy current reads

$$\overrightarrow{J}^+_{1,\ell'} = -\frac{aJ^2}{2}\Big[(1-\sigma^z_{2\ell'-2})\sigma^z_{2\ell'-1}(1-\sigma^z_{2\ell'})\mathbf{D}_{2\ell'-3,2\ell'+1}$$
$$+ (1-\sigma^z_{2\ell'-1})\sigma^z_{2\ell'}(1-\sigma^z_{2\ell'+1})\mathbf{D}_{2\ell'-2,2\ell'+2}$$
$$+ \mathbf{D}_{2\ell'-2,2\ell'+1}\mathbf{K}_{2\ell'-1,2\ell'} + \mathbf{K}_{2\ell'-2,2\ell'+1}\mathbf{D}_{2\ell'-1,2\ell'}\Big], \tag{87}$$

whereas the current of $\mathbf{Q}^-_1$ is given by

$$\overrightarrow{J}^-_{1,\ell'} = -\frac{aJ^2}{2}\Big[(1-\sigma^z_{2\ell'-2})\sigma^z_{2\ell'-1}(1-\sigma^z_{2\ell'})\mathbf{K}_{2\ell'-3,2\ell'+1}$$
$$+ (1-\sigma^z_{2\ell'-1})\sigma^z_{2\ell'}(1-\sigma^z_{2\ell'+1})\mathbf{K}_{2\ell'-2,2\ell'+2}$$
$$+ \mathbf{K}_{2\ell'-2,2\ell'+1}\mathbf{K}_{2\ell'-1,2\ell'} - \mathbf{D}_{2\ell'-2,2\ell'+1}\mathbf{D}_{2\ell'-1,2\ell'} + 4\sigma^z_{2\ell'-1} + 4\sigma^z_{2\ell'} - 4\sigma^z_{2\ell'-1}\sigma^z_{2\ell'}\Big]. \tag{88}$$

Note that the macrosite current equals the spin-site current evaluated at odd sites $\overrightarrow{J}_{\ell'} = \mathbf{j}_{2\ell'-1}$.[9] The continuity equation for $\Delta X = \{\ell'\}$ is represented schematically in Fig. 6. Importantly, having defined the charge densities so as to preserve the configuration, the corresponding currents will commute with $\mathbf{S}^z$, $\mathbf{M}$, as well as with other charges related to the conservation of the configuration.

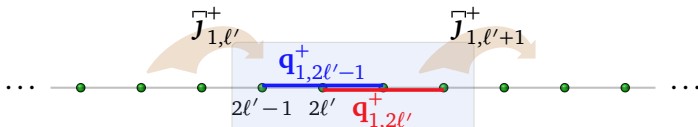

Figure 6: Two macrosites (four neighbouring spins) constitute the unit cell (in light blue), for which continuity equation (85) holds, where the local charge corresponds, for instance, to the Hamiltonian $\mathbf{Q}^+_1 = \tilde{\mathbf{H}}^\eta_F$. On each side of the unit cell another macrosite is coupled to it by the energy current (87) that is supported on six sites.

To the best of our knowledge, the staggered spin along the $z$-axis is the unique strictly local charge that breaks one-site shift invariance. Setting the local density equal to $\overrightarrow{s}^z_{\mathrm{st},\ell'} = (\sigma^z_{2\ell'} - \sigma^z_{2\ell'-1})/2$, the staggered spin current reads

$$\overrightarrow{J}_{\ell'}[\overrightarrow{s}^z_{\mathrm{st}}] = aJ\Big(\mathbf{D}_{2\ell'-2,2\ell'}\frac{1-\sigma^z_{2\ell'-1}}{2} - \mathbf{D}_{2\ell'-3,2\ell'-1}\frac{1-\sigma^z_{2\ell'-2}}{2}\Big) = a(-\mathbf{q}^-_{1,2\ell'-2} + \mathbf{q}^-_{1,2\ell'-3}). \tag{89}$$

We now turn to the current of the quasilocal charge $\mathbf{M}$. As evident from Eq. (16), we can choose the quasilocal operator

$$\overrightarrow{\mathbf{m}}_{\ell'} = \frac{1+\sigma^z_{2\ell'-1}}{2}\sum_{n'=0}^{\frac{L}{2}-1}\left(\prod_{j=2\ell'}^{2\ell'-1+2n'}\frac{1-\sigma^z_j}{2}\right)\frac{1+\sigma^z_{2\ell'+2n'}}{2}, \tag{90}$$

as the charge density. Using $-i[\mathbf{K}, \frac{1+s\sigma^z}{2}\otimes\frac{1+s'\sigma^z}{2}] = \frac{s-s'}{2}\mathbf{D}$, a straightforward calculation gives

$$\overrightarrow{J}_{\ell'}[\overrightarrow{\mathbf{m}}] = \mathbf{D}_{2\ell'-3,2\ell'-1}\frac{1-\sigma^z_{2\ell'-2}}{2}\sum_{n'=0}^{\frac{L}{2}-1}\left(\prod_{j=2\ell'}^{2\ell'-3+2n'}\frac{1-\sigma^z_j}{2}\right)\frac{1+\sigma^z_{2\ell'+2n'-2}}{2}. \tag{91}$$

---

[9]For translationally invariant charges, one can obtain the macrosite continuity equation $\partial_t\overrightarrow{\mathbf{q}}_{\ell'} + \overrightarrow{J}_{\ell'+1} - \overrightarrow{J}_{\ell'} = 0$ by combining spin-site continuity equations $\partial_t\mathbf{q}_{2\ell'-1} + \mathbf{j}_{2\ell'} - \mathbf{j}_{2\ell'-1} = 0$ and $\partial_t\mathbf{q}_{2\ell'} + \mathbf{j}_{2\ell'+1} - \mathbf{j}_{2\ell'} = 0$, whence the identification $\overrightarrow{J}_{\ell'} = \mathbf{j}_{2\ell'-1}$.

## 4.3 Expectation values of charges and currents

### 4.3.1 In a macrostate

In a macrostate characterised by a root density $\rho(p)$, the expectation value of a charge $\mathbf{Q}$ is computed according to Eq. (24), which can be written as

$$\langle \overline{\mathbf{q}}_{\ell'} \rangle - \langle \text{vac}| \overline{\mathbf{q}}_{\ell'} |\text{vac}\rangle = \int_{-\pi}^{\pi} dp \rho(p) q(p). \tag{92}$$

Here, $q(p)$ is the single-particle value of charge $\mathbf{Q}$ and $|\text{vac}\rangle = |\downarrow\downarrow\dots\downarrow\rangle$ is the Bethe Ansatz reference state. On the left-hand side of Eq. (92) we subtract the vacuum expectation value, since the right-hand side is always zero in the vacuum state (the latter contains no momenta, so $\rho_{(\text{vac})}(p) = 0$). The subtraction is allowed, because the charge density expectation value is defined up to an additive constant.

Remarkably, we can also relate the expectation value of the corresponding current to the root density. To that aim, it is convenient to take a step backward and consider first a finite system with $L$ spins. Reference [28] showed that the expectation value of the current in a Bethe state can be expressed in terms of the Gaudin matrix $G$, which is a quantity discernible directly from the Bethe equations, as follows:

$$\langle \overline{J}_{\ell'}[\overline{\mathbf{q}}]\rangle - \langle \text{vac}| \overline{J}_{\ell'}[\overline{\mathbf{q}}] |\text{vac}\rangle = \sum_{i,j=1}^{N} E'(p_i)[G^{-1}]_{ij} q(p_j). \tag{93}$$

Here, $(\bullet)'$ denotes the derivative, and $N$ is fixed because both the charge density and the current commute with the total magnetisation along the $z$-axis, $\mathbf{S}^z$. The elements of the Gaudin matrix are given by

$$G_{ij} = \partial_{p_j}(2\pi J_i). \tag{94}$$

Analogously to what was said for $N$, the partial derivative in Eq. (94) is taken at fixed $M$. Indeed, both the charge density and the current commute with $\mathbf{M}$ as well, and hence the continuity equation (85) can be enforced at fixed $M$. Comparing Eqs (21), (78), and (94), we see that the Gaudin matrix corresponds to the Jacobian of the transformation $p \mapsto \wp$, up to a multiplicative factor. In the end we find

$$G_{ij} = \left(\frac{L}{2} + M\right)\delta_{i,j} - \frac{M}{N}. \tag{95}$$

Using $E(p) = 4J \cos p$, the expectation value of the current then reads

$$\langle \overline{J}_{\ell'}[\overline{\mathbf{q}}]\rangle - \langle \text{vac}| \overline{J}_{\ell'}[\overline{\mathbf{q}}] |\text{vac}\rangle = -\frac{4J}{\frac{L}{2}+M}\sum_{i=1}^{N} q(p_i)\sin p_i - \frac{8MJ}{LN(\frac{L}{2}+M)}\sum_{i=1}^{N}\sin p_i \sum_{i=1}^{N} q(p_i). \tag{96}$$

Relation (96) is expected to hold for the currents of all local charges $\mathbf{Q}_n^{\pm}$,[10] and we have checked it for $\mathbf{J}_1^{\pm}$ (see Eqs (87) and (88)) by means of exact diagonalisation, up to $L = 14$ sites. In the thermodynamic limit it becomes

$$\langle \overline{J}_{\ell'}[\overline{\mathbf{q}}]\rangle - \langle \text{vac}| \overline{J}_{\ell'}[\overline{\mathbf{q}}] |\text{vac}\rangle \xrightarrow{\text{TD}} \int_{-\pi}^{\pi} dp \rho(p) v(p) q(p), \tag{97}$$

---

[10]The expectation value of $\mathbf{M}$'s current will be derived in the next section directly in the thermodynamic limit.

providing the sought-after relation between the expectation value of the current and the root density. Here, $v(p) = \partial_p \varepsilon(p)/\partial_p \wp(p)$ is the effective velocity of quasiparticles, explicitly expressible as

$$v(p) = E'(p) + \mu \int_{-\pi}^{\pi} \frac{\mathrm{d}k}{2\pi} n(k)[E'(k) - E'(p)] = \frac{E'(p) + \mu \langle E' \rangle}{1 + \mu \langle 1 \rangle}, \tag{98}$$

where we used the dressing equations (78) and (79).

Finally, summing Eq. (96) over the macrosites gives the expectation value of the total current. Its diagonal (stationary) part, which we indicate with $\mathbf{J}_{\mathrm{d}}[\bullet]$, can be readily extracted and reads

$$\mathbf{J}_{\mathrm{d}}[\mathbf{Z}_n] = iJ \frac{L}{\frac{L}{2} + \mathbf{M}} (\mathbf{Z}_{n+1} - \mathbf{Z}_{n-1}) + 2iJ \frac{\mathbf{M}}{\mathbf{Z}_0} \frac{\mathbf{Z}_1 - \mathbf{Z}_1^{\dagger}}{\frac{L}{2} + \mathbf{M}} \mathbf{Z}_n, \tag{99}$$

where $\mathbf{Z}_n(\equiv \mathbf{Z}_{-n}^{\dagger}) = \frac{1}{4J}(\mathbf{Q}_n^+ + i\mathbf{Q}_n^-)$. In the noninteracting sector, where $\mathbf{M} = 0$, $\mathbf{J}_{\mathrm{d}}[\mathbf{Z}_n]$ are equivalent to local operators. This property is, however, immediately lost in the presence of interactions.[11]

### 4.3.2 In a locally quasistationary state

Reference [26] named "locally quasistationary state" (LQSS) the inhomogeneous macrostate that captures the expectation values of local operators in an inhomogeneous out-of-equilibrium state at asymptotically large times, after the fastest degrees of freedom have stopped affecting the dynamics of local observables. In particular, at each point in space an LQSS is locally equivalent to some stationary state. A route to a formal definition of locally quasistationary states is based on the identification of the smallest families $\mathcal{C}$ of operators, including both the set of conserved charges and their densities, that are closed under time evolution [25]. A locally quasistationary state could than be represented by a density matrix $\boldsymbol{\rho} = e^{\mathbf{W}}/\mathrm{tr}[e^{\mathbf{W}}]$, with $\mathbf{W} \in \mathcal{C}$. A basic property of the family $\mathcal{C}$ is that, for any conserved operator $\mathbf{O} \in \mathcal{C}$, its charge density is defined in such a way that $\mathbf{J}_{\mathrm{d}}[\mathbf{O}]$ (which generalises functional (99) to the conserved operators belonging to $\mathcal{C}$) is in $\mathcal{C}$ as well. Notwithstanding the simplicity of Eq. (99) suggesting the potential to identify a family $\mathcal{C}$ of such operators in the folded XXZ model, we leave this problem to future investigation. Here, we stick to the heuristic method that has been used to obtain the first-order generalised hydrodynamics (GHD) equation in interacting integrable systems.

To that aim, we consider the continuity equation (85) in the limit described by Eq. (86):

$$\partial_t \langle \mathbf{q} \rangle (x, t) + \partial_x \langle \mathbf{j} \rangle (x, t) = 0. \tag{100}$$

The local equivalence of an LQSS to a stationary state indicates the possibility to lift the root density to a function of space and time. Let us then assume that the expectation values in the LQSS characterised by the density matrix $e^{\mathbf{W}}/\mathrm{tr}[e^{\mathbf{W}}]$, with $\mathbf{W} \in \mathcal{C}$, are completely determined by a root density depending on space and time. Even if we do not really know how to compute expectation values in such an inhomogeneous macrostate, if the range of the operator is much

---

[11]Note that, as discussed in the first part of our work [1], for finite $L$ the local charges $\mathbf{Z}_n$ are not functionally independent. Indeed, we have

$$\mathbf{Z}_0 \mathbf{Z}_{\frac{L}{2}+\ell} = \sum_{M=0}^{\frac{L}{2}} \mathbf{\Pi}_M \mathbf{Z}_{\ell-M} \mathbf{Z}_{\frac{L}{2}+M},$$

where $\mathbf{\Pi}_M$ is the projector on the subspace with $\mathbf{M} = M$ (see Ref. [29] for some observations concerning the independence of integrals of motion in the presence of such a functional dependence). This is a further manifestation of the incompleteness of the charges $\mathbf{Q}_n$.

smaller than the typical length of the inhomogeneity, we can simply compute its expectation value in the macrostate characterised by $\rho_{x,t}(p)$, by treating $x$ and $t$ as external parameters. Under such a low-inhomogeneity assumption, using Eqs (92) and (97), we finally obtain

$$\int_{-\pi}^{\pi} dp \big( \partial_t \rho_{x,t}(p) + \partial_x[\rho_{x,t}(p)v_{x,t}(p)] \big) q(p) = \mathcal{O}(\partial_x^2). \tag{101}$$

The higher order derivatives signify the corrections due to the modulation of the root density on length scales smaller than the typical size of the inhomogeneity. Since the single-particle values $q_n^\pm(p)$ of the charges $\mathbf{Q}_n^\pm$ form a basis in the space of square-integrable functions of the momentum $p \in [-\pi, \pi)$, we end up with the fundamental equation of generalised hydrodynamics

$$\boxed{\partial_t \rho_{x,t}(p) + \partial_x[\rho_{x,t}(p)v_{x,t}(p)] = \mathcal{O}(\partial_x^2).} \tag{102}$$

By symmetry, the density of holes $\rho_h(p)$ is expected to satisfy the same equation, which then implies

$$\partial_t \rho_{t;x,t} + \partial_x[\rho_{t;x,t} v_{x,t}(p)] = \mathcal{O}(\partial_x^2), \tag{103}$$

where we used that, in our specific case, $\rho_t = \rho(p) + \rho_h(p)$ does not depend on the rapidity.

At first sight the latter equation might look wrong, since it relates the rapidity-independent function $\rho_t$ to a function that, instead, seemingly depends on the rapidity. This is only apparent: plugging Eq. (98) into the continuity equation (103) and using $2\pi\rho_t = 1 + \mu\langle 1 \rangle$ (cf. Eq. (26)), we obtain

$$\partial_t \rho_{t;x,t} + \partial_x \left( \frac{\mu_{x,t}}{2\pi} \langle E' \rangle_{x,t} \right) = \mathcal{O}(\partial_x^2), \tag{104}$$

where we used notation (77). By virtue of Eq. (102), this can be recast into a continuity equation for $\mu_{x,t}$:

$$\boxed{\partial_t \mu_{x,t} + \frac{\langle E' \rangle_{x,t}}{\langle 1 \rangle_{x,t}} \partial_x \mu_{x,t} = \mathcal{O}(\partial_x^2).} \tag{105}$$

Equations (102) and (103) can also be used to obtain the dynamical equation satisfied by the filling function $n_{x,t}(p) = \rho_{x,t}(p)/\rho_{t;x,t}$, which is given by

$$\boxed{\partial_t n_{x,t}(p) + v_{x,t}(p) \partial_x n_{x,t}(p) = \mathcal{O}(\partial_x^2).} \tag{106}$$

The $\mathcal{O}(\partial_x^2)$ corrections become irrelevant in a particular scaling limit, often refered to as "Euler scale" or "ballistic scaling limit". In this limit one can truncate generalised hydrodynamics at the first order. The description of diffusive, sub-diffusive, and Tracy-Widom scaling behaviours requires the development of higher-order generalised hydrodynamics [25, 30, 31].

Finally, we note $\frac{2}{L}\langle \mathbf{M} \rangle_{x,t} = 2\pi\rho_{t;x,t} - 1$, therefore Eq. (104) can be used to infer the expectation value of the corresponding current $\overline{\mathbf{j}}_{\ell'}[\overline{\mathbf{m}}]$, given by Eq. (91), in a macro-state:

$$\langle \overline{\mathbf{j}}_{\ell'}[\overline{\mathbf{m}}] \rangle - \langle \text{vac} | \overline{\mathbf{j}}_{\ell'}[\overline{\mathbf{m}}] | \text{vac} \rangle = \mu \int_{-\pi}^{\pi} dp \rho(p) E'(p). \tag{107}$$

In other words, the diagonal part of the total current reads

$$\mathbf{J}_d[\mathbf{M}] = 2iJ \frac{\mathbf{M}}{\mathbf{Z}_0} (\mathbf{Z}_1 - \mathbf{Z}_1^\dagger). \tag{108}$$

An attentive reader might have noticed that the picture developed so far does not include the staggered magnetisation. We *conjecture* that the staggered magnetisation – as well as the

other quasilocal charges whose eigenvalues describe the configuration – satisfies the same GHD equation as $\mu = \langle \overline{\mathbf{m}} \rangle / \langle 1 \rangle$, namely

$$\partial_t \left( \frac{m^z_{\text{st};x,t}}{\langle 1 \rangle_{x,t}} \right) + \frac{\langle E' \rangle_{x,t}}{\langle 1 \rangle_{x,t}} \partial_x \left( \frac{m^z_{\text{st};x,t}}{\langle 1 \rangle_{x,t}} \right) = \mathcal{O}(\partial_x^2). \tag{109}$$

In the next section we study the first-order GHD equation, expressed by Eqs (105) and (106), in the inhomogeneous setting where two local macrostates are joined together.

# 5 Junction of two local macrostates

A nonequilibrium setting that has been intensively studied for its potential to elucidate transport properties of both classical and quantum many-body systems is the junction of two pieces of materials kept at different temperatures [32–39]. The system of interest is typically sandwiched between the materials at the junction. In this case the materials act as thermal baths, and such a setting is a paradigm of a non-isolated system. If, however, the two pieces are in direct contact, one can take both of them as constituents of a larger isolated system. In the latter, however big the pieces of materials are, they will experience a nontrivial evolution over time. In some cases the evolution can be captured by assuming local thermalisation, where the state of the system is still characterised by a temperature, but the latter depends on space and time. An important exception are the integrable systems, where local thermal equilibrium states do not form invariant subspaces. In other words, the nonequilibrium state can not be characterised solely by the time evolution of the local temperature. Instead, the full set of generalised chemical potentials (or generalised temperatures) is required for the determination of such an out-of-equilibrium state.

We consider here the junction of two local macrostates in the folded XXZ model, for example, two thermal states. Besides its physical relevance, this scenario carries some mathematical simplifications. Firstly, the boundary conditions, i.e., $n(p)$ and $\mu$ in the initial state, are functions of the ray $\zeta = x/t$, so the solution to the GHD equations is expected to depend on $x$ and $t$ only through their ratio. Secondly, the corrections to Eqs (105) and (106) are irrelevant in the ballistic (B) scaling limit $t \to \infty$, at fixed $\zeta = x/t$, so the GHD equations reduce to

$$(\zeta - v_\zeta(p))\partial_\zeta n_\zeta(p) \xrightarrow{\text{B}} 0, \qquad (\zeta - V_\zeta)\partial_\zeta \mu_\zeta \xrightarrow{\text{B}} 0, \tag{110}$$

where we have defined

$$V_\zeta := \frac{\langle E' \rangle_\zeta}{\langle 1 \rangle_\zeta} = \frac{\int_{-\pi}^{\pi} \mathrm{d}p \, n_\zeta(p) E'(p)}{\int_{-\pi}^{\pi} \mathrm{d}p \, n_\zeta(p)} . \tag{111}$$

The ballistic scaling limit is supposed to exist whenever the limits

$$n_\pm(p) = \lim_{x \to \pm\infty} n_{x,0}(p), \qquad \mu_\pm = \lim_{x \to \pm\infty} \mu_{x,0}, \tag{112}$$

exist.

## 5.1 On the solution to the GHD equation

The initial conditions (112) fix the values of $n_\zeta(p)$ and $\mu_\zeta$ in the limits $\zeta \to \pm\infty$. Thus, in order for the solution to be unique, the equations $\zeta = v_\zeta(p)$ (for each given momentum $p$) and $\zeta = V_\zeta$ should only have one solution. In this section such uniqueness conditions will just be assumed to be satisfied.

The solution to the second of the GHD equations (110), namely, the equation for $\mu_\zeta$, is a piecewise constant function with one discontinuity (due to the uniqueness assumption). It is then convenient to define the auxiliary velocities

$$
\begin{aligned}
v_\zeta^\pm(p) &:= E'(p) + \mu_\pm \int_{-\pi}^{\pi} \frac{\mathrm{d}k}{2\pi} n_\zeta^\pm(k)[E'(k)-E'(p)] = \frac{E'(p)+\mu_\pm \langle E'\rangle_\zeta^\pm}{1+\mu_\pm \langle 1\rangle_\zeta^\pm}, \\
V_\zeta^\pm &:= \frac{\langle E'\rangle_\zeta^\pm}{\langle 1\rangle_\zeta^\pm} = \frac{\int_{-\pi}^{\pi} \mathrm{d}p\, n_\zeta^\pm(p)E'(p)}{\int_{-\pi}^{\pi} \mathrm{d}p\, n_\zeta^\pm(p)},
\end{aligned}
\tag{113}
$$

and the auxiliary GHD equations

$$
(\zeta - v_\zeta^\pm(p))\partial_\zeta n_\zeta^\pm(p) = 0, \qquad (\zeta - V_\zeta^\pm)\partial_\zeta \mu_\zeta^\pm = 0,
\tag{114}
$$

with the same boundary conditions as in the original problem. The sign in the superscript or subscript refers to the region, with respect to the discontinuity in $\mu_\zeta$, where the equations or quantities are defined.

The full solution to Eq. (110), with a single discontinuity, can be constructed from the solutions to the auxiliary GHD equations (114) by joining $n_\zeta^-(p)$ and $n_\zeta^+(p)$ at the ray of the discontinuity in $\mu_\zeta$, i.e., the ray that solves both $\zeta = V_\zeta^-$ and $\zeta = V_\zeta^+$. The latter two equations should therefore have the same solution. To find it, we note that a change in $\mu_+$ does not affect $V_\zeta^-$. This implies that the common solution of equations $\zeta = V_\zeta^-$ and $\zeta = V_\zeta^+$ is, in fact, independent of $\mu_+$. We can then specialise the calculation to the noninteracting case $\mu_+ = 0$, in which the effective velocity is simply $v^+(p) = E'(p) = -4J\sin p$, so that $n_\zeta^+(p) = n_{\mathrm{sgn}(4J\sin p + \zeta)}(p)$. The equation $\zeta = V_\zeta^+$ can then be rewritten as

$$
0 = f(\zeta; n_+(p), n_-(p)) := \int_{-\pi}^{\pi} \mathrm{d}p\, n_{\mathrm{sgn}(4J\sin p + \zeta)}(p)\left(\sin p + \frac{\zeta}{4J}\right).
\tag{115}
$$

Since $f(\zeta; n_+(p), n_-(p))$ is a monotonous function of $\zeta$, and $\lim_{\zeta\to\pm\infty} f(\zeta; n_+(p), n_-(p)) = \pm\infty$, equation $f(\zeta; n_+(p), n_-(p)) = 0$ has a single solution $\zeta = \zeta_\mathrm{d}$. This simple observation allows us to conclude that, assuming a single discontinuity in $\mu$, the ray $\zeta_\mathrm{d}$ of the discontinuity is the unique zero of $f(\zeta; n_+(p), n_-(p))$.

Again assuming uniqueness, the first of the GHD equations (110) is solved by

$$
n_\zeta(p) = \frac{n_\zeta^+(p)+n_\zeta^-(p)}{2} + \frac{n_\zeta^+(p)-n_\zeta^-(p)}{2}\mathrm{sgn}(\zeta - \zeta_\mathrm{d}),
\tag{116}
$$

where $n_\zeta^\pm(p)$ are the filling functions on each side of the ray of the discontinuity. They separately solve the first of the auxiliary GHD equations (114) and read

$$
n_\zeta^\pm(p) = \frac{n_+(p)+n_-(p)}{2} + \frac{n_+(p)-n_-(p)}{2}\mathrm{sgn}(\zeta - v_\zeta^\pm(p)).
\tag{117}
$$

In addition to this solution, the following two facts should be kept in mind:

1. The maximal (minimal) ray at which the filling function undergoes a transition from $n_-(p)$ to $n_+(p)$ corresponds to the right (left) edge of the light cone, emanating at the junction of the two local macrostates. It is given by

$$
\zeta_\mathrm{lc}^\pm = \frac{\pm 4J + \mu_\pm \langle E'\rangle_\pm}{1+\mu_\pm \langle 1\rangle_\pm}.
\tag{118}
$$

   In the ballistic scaling limit, the expectation value of any local observable outside the light cone is constant and equal to its initial value.

2. At the ray $\zeta = \zeta_{\mathrm{d}}$ of the discontinuity we have

$$\zeta_{\mathrm{d}} - v_{\zeta_{\mathrm{d}}}^{\pm}(p) = \frac{\zeta_{\mathrm{d}} - E'(p)}{1 + \mu_{\pm}\langle 1\rangle_{\zeta_{\mathrm{d}}}^{\pm}}, \tag{119}$$

where we have used $\zeta_{\mathrm{d}} = V_{\zeta_{\mathrm{d}}}^{\pm}$ and the definitions (113) of the auxiliary velocities. Since $1 + \mu_{\pm}\langle 1\rangle_{\zeta_{\mathrm{d}}}^{\pm} \geq 0$, this implies

$$\mathrm{sgn}(\zeta_{\mathrm{d}} - v_{\zeta_{\mathrm{d}}}^{+}(p)) = \mathrm{sgn}(\zeta_{\mathrm{d}} - v_{\zeta_{\mathrm{d}}}^{-}(p)) = \mathrm{sgn}(\zeta_{\mathrm{d}} - E'(p)), \tag{120}$$

for all momenta $p$. In particular, for momenta $p$ that satisfy $\zeta_{\mathrm{d}} = E'(p) = -4J\sin p$, the filling function changes its value from $n_-(p)$ to $n_+(p)$ at the ray $\zeta_{\mathrm{d}}$. This will be used in the next section, where we discuss the discontinuities in the expectation values of local charges.

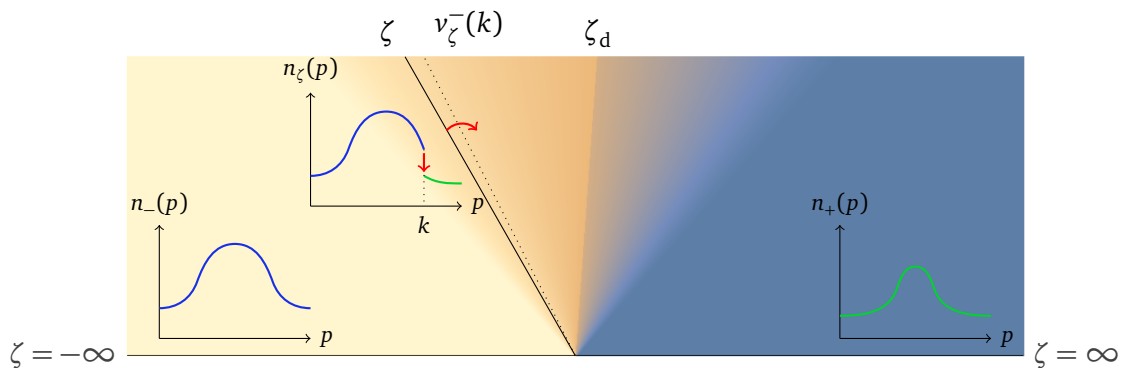

Figure 7: When the ray $\zeta$ crosses $v_{\zeta}^{-}(k)$, the value of the filling function at momentum $k$ jumps from $n_-(k)$ to $n_+(k)$. When $\zeta_{\mathrm{d}}$ is crossed, $\mu_-$ changes discontinuously into $\mu_+$, and $v_{\zeta}^{-}(k)$, at $k$ satisfying $E'(k) = \zeta_{\mathrm{d}}$, continuously into $v_{\zeta}^{+}(k)$. Sweeping the interval $\zeta \in (-\infty, \infty)$, the left-hand side state $n_-$ completely transforms into $n_+$. The data shown corresponds to the junction of two thermal states with $\beta_- = 0.25$ and $\beta_+ = 2.75$. Clearly visible are the asymmetric light cone between the rays $\zeta_{\mathrm{lc}}^{-} \approx -3.0140J$ and $\zeta_{\mathrm{lc}}^{+} \approx 2.9400J$, computed according to formula (118), as well as the ray of discontinuity $\zeta_{\mathrm{d}} \approx 0.2416J$. The sketched filling functions are purely qualitative.

To summarise, the picture that unfolds is as follows (see Fig. 7). When the ray $\zeta$ sweeps from $\zeta = \zeta_{\mathrm{lc}}^{-}$ to $\zeta = \zeta_{\mathrm{d}}$, the filling function transforms from $n_-(p)$ to $n_+(p)$ at the momenta $p$, for which $\zeta = v_{\zeta}^{-}(p)$. When $\zeta_{\mathrm{d}}$ is crossed, $v_{\zeta}^{-}(p)$ is replaced by $v_{\zeta}^{+}(p)$, the change being, in fact, continuous. By "continuous" we mean that, at the momenta $p$ for which the transition of the filling function from $n_-(p)$ to $n_+(p)$ occurs at $\zeta_{\mathrm{d}}$ (these momenta satisfy $\zeta_{\mathrm{d}} = E'(p)$), the velocities coincide: $v_{\zeta_{\mathrm{d}}}^{+}(p) = v_{\zeta_{\mathrm{d}}}^{-}(p) = \zeta_{\mathrm{d}}$.

## 5.2 Non-ballistic behaviour

Let us assume that the filling functions of the boundary conditions are smooth; in this way any non-analytic behaviour can be traced back to the "domain-wall" structure of the initial state. As it generally happens, the discontinuity in the filling functions that is caused by the discontinuity of the initial state does not directly produce non-analytic behaviour in the profiles

of the local observables (except for the neighbourhoods of the light cones). Indeed, if $\mu_+ = \mu_-$, the ballistic profiles are smooth inside the light cone[12]. On the other hand, a discontinuity in $\mu$ at the initial time immediately spoils the smoothness of the profiles. At first sight, this bears resemblance to the effect of a discontinuous sign of magnetisation in the XXZ model [40]. However, in that case, in the ballistic scaling limit the operators that are even under the change in the sign of $\mathbf{S}^z$ are not affected by the discontinuity. In contrast, in our case $\mu_+ \neq \mu_-$ makes the ballistic profile of *any* generic local charge and current discontinuous. In other words, there is no global symmetry (independent of the initial state) that protects the smoothness of the profiles. In the following we quantify the discontinuities of charges and currents, which turn out to be deducible a priori.

Denoting $f_{\zeta_d^-} = \lim_{\zeta \nearrow \zeta_d} f_\zeta$ and $f_{\zeta_d^+} = \lim_{\zeta \searrow \zeta_d} f_\zeta$, the left and right limits of the expectation values of charges and currents at the discontinuity are given by

$$
\begin{aligned}
\langle \overline{\mathbf{q}}_{\ell'} \rangle_{\zeta_d^\pm} &= \int_{-\pi}^{\pi} \mathrm{d}p \, \rho_{\zeta_d^\pm}(p) q(p) = \frac{\int_{-\pi}^{\pi} \mathrm{d}p \, n_{\zeta_d}^\pm(p) q(p)}{2\pi - \mu_\pm \int_{-\pi}^{\pi} \mathrm{d}p \, n_{\zeta_d}^\pm(p)}, \\
\langle \overline{J}_{\ell'}[\overline{\mathbf{q}}] \rangle_{\zeta_d^\pm} &= \int_{-\pi}^{\pi} \mathrm{d}p \, \rho_{\zeta_d^\pm}(p) v_{\zeta_d^\pm}(p) q(p) = \frac{\int_{-\pi}^{\pi} \mathrm{d}p \, n_{\zeta_d}^\pm(p) v_{\zeta_d}^\pm(p) q(p)}{2\pi - \mu_\pm \int_{-\pi}^{\pi} \mathrm{d}p \, n_\zeta^\pm(p)},
\end{aligned}
\tag{121}
$$

where $n_{\zeta_d}^\pm(p)$ and $v_{\zeta_d}^\pm(p)$ readily follow from $\zeta_d = V_{\zeta_d}^\pm$ and Eqs (113), (117), and (120):

$$
\begin{aligned}
n_{\zeta_d}^\pm(p) &= n_{\mathrm{sgn}(\zeta_d + 4J\sin p)}(p) \\
v_{\zeta_d}^\pm(p) &= -4J\sin p + \mu_\pm(\zeta_d + 4J\sin(p)) \int_{-\pi}^{\pi} \frac{\mathrm{d}k}{2\pi} n_{\mathrm{sgn}(\zeta_d + 4J\sin k)}(k).
\end{aligned}
\tag{122}
$$

Remarkably, these expressions, along with Eq. (115), are written solely in terms of the boundary conditions. We also note that the existence of two rays $\zeta_d^\pm$, described by the same filling function but different $\mu$, rules out the possibility to interpret the locally quasistationary state as a collection of local macrostates, since in the latter $\mu$ is fixed by Eq. (39). In particular, since thermal states at different temperatures have different values of $\mu$ (see Fig. 3), this observation applies also to the junction of thermal states.

The charge discontinuity at ray $\zeta = \zeta_d$ is now given by

$$
\langle \overline{\mathbf{q}} \rangle_{\zeta_d^+} - \langle \overline{\mathbf{q}} \rangle_{\zeta_d^-} = \frac{(\mu_+ - \mu_-)\int_{-\pi}^{\pi} \mathrm{d}k \, n_{\mathrm{sgn}(\zeta_d + 4J\sin k)}(k) \int_{-\pi}^{\pi} \mathrm{d}p \, n_{\mathrm{sgn}(\zeta_d + 4J\sin p)}(p) q(p)}{(2\pi - \mu_+ \int_{-\pi}^{\pi} \mathrm{d}k \, n_{\mathrm{sgn}(\zeta_d + 4J\sin k)}(k))(2\pi - \mu_- \int_{-\pi}^{\pi} \mathrm{d}p \, n_{\mathrm{sgn}(\zeta_d + 4J\sin p)}(p))},
\tag{123}
$$

which implies

$$
\frac{\langle \overline{\mathbf{q}} \rangle_{\zeta_d^+} - \langle \overline{\mathbf{q}} \rangle_{\zeta_d^-}}{\langle \overline{\mathbf{q}} \rangle_{\zeta_d^\pm}} = (\mu_+ - \mu_-) \langle 1 \rangle_{\zeta_d^\mp}.
\tag{124}
$$

The current discontinuity at ray $\zeta = \zeta_d$ instead reads

$$
\langle \overline{J}[\overline{\mathbf{q}}] \rangle_{\zeta_d^+} - \langle \overline{J}[\overline{\mathbf{q}}] \rangle_{\zeta_d^-} = \zeta_d (\langle \overline{\mathbf{q}} \rangle_{\zeta_d^+} - \langle \overline{\mathbf{q}} \rangle_{\zeta_d^-}),
\tag{125}
$$

which can be immediately inferred from the continuity equation $\zeta \, \partial_\zeta \langle \overline{\mathbf{q}} \rangle - \partial_\zeta \langle \overline{J}[\overline{\mathbf{q}}] \rangle = 0$.

---

[12]We remind the reader that in the XXZ model the presence of (for $\Delta \geq 1$, infinitely) many species of excitations creates non-analyticities at the light cones of the species [40].

Remarkably, the ratio of expectation values of any two local charge densities $\overline{\mathbf{q}}_a$ and $\overline{\mathbf{q}}_b$ is continuous

$$\frac{\langle \overline{\mathbf{q}}_a \rangle_{\zeta_d^{\pm}}}{\langle \overline{\mathbf{q}}_b \rangle_{\zeta_d^{\pm}}} = \frac{\int_{-\pi}^{\pi} dp \, n_{\mathrm{sgn}(\zeta_d + 4J \sin p)}(p) q_a(p)}{\int_{-\pi}^{\pi} dp \, n_{\mathrm{sgn}(\zeta_d + 4J \sin p)}(p) q_b(p)} . \tag{126}$$

Although this does not extend to the expectation values of currents, it holds true for the shifted currents $\overline{\mathbf{J}}[\overline{\mathbf{q}}] + \overline{\mathbf{q}} \star \overline{\mathbf{q}}_1^-$, where $\overline{\mathbf{q}}_a \star \overline{\mathbf{q}}_b$ denotes the charge with a single-particle value $q_a(p) q_b(p)$. Indeed, we have

$$\frac{\langle \overline{\mathbf{J}}[\overline{\mathbf{q}}_a] + \overline{\mathbf{q}}_a \star \overline{\mathbf{q}}_1^- \rangle_{\zeta_d^{\pm}}}{\langle \overline{\mathbf{J}}[\overline{\mathbf{q}}_b] + \overline{\mathbf{q}}_b \star \overline{\mathbf{q}}_1^- \rangle_{\zeta_d^{\pm}}} = \frac{\int_{-\pi}^{\pi} dp \, n_{\mathrm{sgn}(\zeta_d + 4J \sin p)}(p)(\zeta_d + 4J \sin p) q_a(p)}{\int_{-\pi}^{\pi} dp \, n_{\mathrm{sgn}(\zeta_d + 4J \sin p)}(p)(\zeta_d + 4J \sin p) q_b(p)} . \tag{127}$$

Similar calculations can be performed for the expectation value of $\overline{\mathbf{m}}$. Here, the left and right limits are given by

$$\langle \overline{\mathbf{m}} \rangle_{\zeta_d^{\pm}} = \mu_{\pm} \langle 1 \rangle_{\zeta_d^{\pm}} = \frac{\mu_{\pm} \int_{-\pi}^{\pi} dp \, n_{\mathrm{sgn}(\zeta_d + 4J \sin p)}(p)}{2\pi - \mu_{\pm} \int_{-\pi}^{\pi} dp \, n_{\mathrm{sgn}(\zeta_d + 4J \sin p)}(p)} , \tag{128}$$

while the discontinuity reads

$$\langle \overline{\mathbf{m}} \rangle_{\zeta_d^{+}} - \langle \overline{\mathbf{m}} \rangle_{\zeta_d^{-}} = \frac{2\pi(\mu_{+} - \mu_{-}) \int_{-\pi}^{\pi} dp \, n_{\mathrm{sgn}(\zeta_d + 4J \sin p)}(p)}{(2\pi - \mu_{+} \int_{-\pi}^{\pi} dp \, n_{\mathrm{sgn}(\zeta_d + 4J \sin p)}(p))(2\pi - \mu_{-} \int_{-\pi}^{\pi} dp \, n_{\mathrm{sgn}(\zeta_d + 4J \sin p)}(p))} . \tag{129}$$

Finally, we point out that the GHD equation for $\mu$ can be replaced by the additional boundary condition $n_{\zeta_d}(p) = n_{\mathrm{sgn}(\zeta_d + 4J \sin p)}(p)$ for the filling function in the two independent regions $\zeta < \zeta_d$ and $\zeta > \zeta_d$, in which $\mu$ is constant.

**Entanglement entropy**

The expectation values of charges and currents are not the only physical quantities that exhibit non-ballistic behaviour. At large time, the entanglement entropy of a subsystem per unit length is supposed to approach the Yang-Yang entropy density in the emerging locally quasistationary state [41, 42]. In our case the Yang-Yang entropy is reported in Eq. (33).[13] For the sake of simplicity we assume that the initial state has zero staggered magnetisation. Then, around the discontinuity, the Yang-Yang entropy reads

$$s_{\mathrm{YY};\zeta_d^{\pm}} = \frac{H(2\mu_{\pm}) \int_{-\pi}^{\pi} dp \, n_{\mathrm{sgn}(\zeta_d + 4J \sin p)}(p) + \int_{-\pi}^{\pi} dp \, H(n_{\mathrm{sgn}(\zeta_d + 4J \sin p)}(p))}{2\pi - \mu_{\pm} \int_{-\pi}^{\pi} dp \, n_{\mathrm{sgn}(\zeta_d + 4J \sin p)}(p)} . \tag{130}$$

The resulting discontinuity is computed as

$$s_{\mathrm{YY};\zeta_d^{+}} - s_{\mathrm{YY};\zeta_d^{-}} = \frac{(\mu_{+} - \mu_{-}) \int_{-\pi}^{\pi} dp \, H(n_{\mathrm{sgn}(\zeta_d + 4J \sin p)}(p)) \int_{-\pi}^{\pi} dp \, n_{\mathrm{sgn}(\zeta_d + 4J \sin p)}(p)}{(2\pi - \mu_{+} \int_{-\pi}^{\pi} dp \, n_{\mathrm{sgn}(\zeta_d + 4J \sin p)}(p))(2\pi - \mu_{-} \int_{-\pi}^{\pi} dp \, n_{\mathrm{sgn}(\zeta_d + 4J \sin p)}(p))} +$$
$$+ \frac{H(2\mu_{+}) \int_{-\pi}^{\pi} dp \, n_{\mathrm{sgn}(\zeta_d + 4J \sin p)}(p)}{2\pi - \mu_{+} \int_{-\pi}^{\pi} dp \, n_{\mathrm{sgn}(\zeta_d + 4J \sin p)}(p)} - \frac{H(2\mu_{-}) \int_{-\pi}^{\pi} dp \, n_{\mathrm{sgn}(\zeta_d + 4J \sin p)}(p)}{2\pi - \mu_{-} \int_{-\pi}^{\pi} dp \, n_{\mathrm{sgn}(\zeta_d + 4J \sin p)}(p)} . \tag{131}$$

---

[13]We are assuming that the locally quasistationary state is locally equivalent to a quasilocal macrostate defined in Section 2.2.2. If that is not the case, our estimation of the Yang-Yang entropy becomes an upper bound.

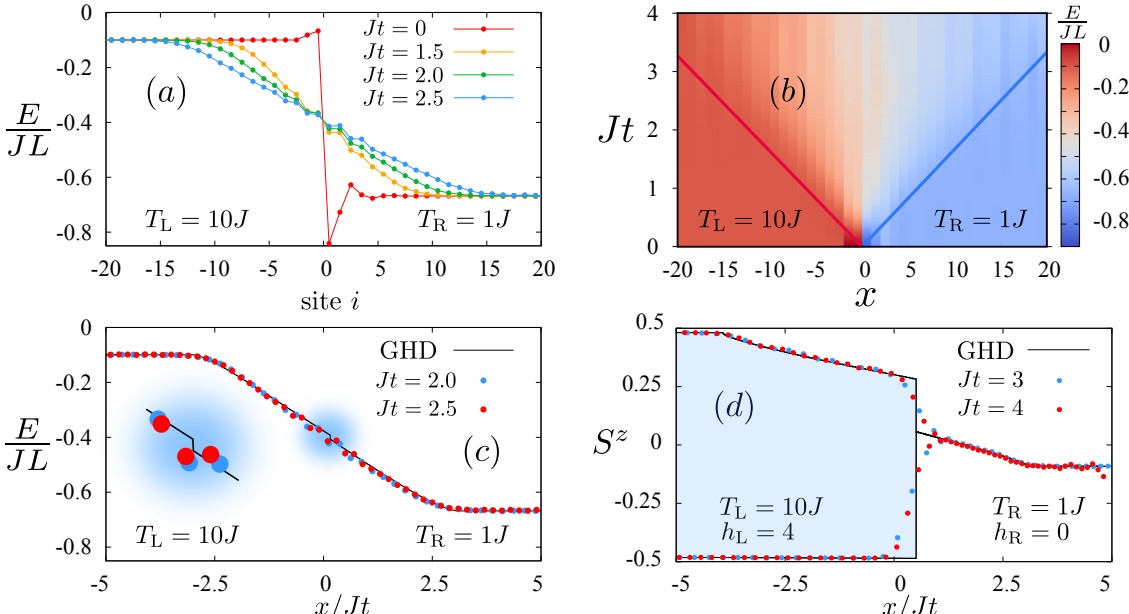

Figure 8: Panels (a) and (c) show evolution of the energy profile after a sudden junction of two thermal states. Red and blue lines on panel (b) denote the light cone rays $\zeta_{\text{lc}}^- = -6.00495J$ and $\zeta_{\text{lc}}^+ = 6.12411J$, respectively, computed from Eq. (118). The animated version of the energy profile is accessible through QR codes in Fig. 9. Panels (c) and (d) show comparison between the numerically computed profiles of energy (c) and magnetisation (d) on the one hand (both *per spin site*), and the GHD prediction on the other. In panel (c), the GHD prediction for the macrosite energy density has been divided by 2, in order to obtain the average energy per spin site. In panel (d), the magnetisation on the left-hand side oscillates between the solid lines that border the shaded region (it is staggered as a result of the initial condition). The shaded region expands in time, as the discontinuity propagates towards the right-hand side with velocity computed from Eq. (115). In all plots, the profiles are centered at the geometrical center of the spin chain used in the simulation.

As a case study we could consider the junction of the ground state, in which $\mu_- = 1/2$, $n_-(p) = \theta(|p| - k_{\text{F}})$, with a pure state in which any traceless charge has zero expectation value: the late-time behaviour on the right of the light cone would then be described by the infinite temperature state, where $\mu_+ = 1/3$, $n_+(p) = 3/4$. Calculation then yields $\zeta_{\text{d}} \approx -0.272252J$, $\int_{-\pi}^{\pi} \text{d}p\, n_{\text{sgn}(\zeta_{\text{d}} + 4J \sin p)}(p) \approx 4.16302$, and $\int_{-\pi}^{\pi} \text{d}p\, H(n_{\text{sgn}(\zeta_{\text{d}} + 4J \sin p)}(p)) \approx 1.72832$. The entropy discontinuity is then given by $s_{\text{YY};\zeta_{\text{d}}^+} - s_{\text{YY};\zeta_{\text{d}}^-} \approx 0.482976$. To the best of our knowledge, this is the first model where a discontinuity is predicted in the ballistic scaling limit of the entanglement entropy.

## 5.3 Comparison with numerical simulations

Figure 8 shows a comparison between the generalised-hydrodynamics predictions and the numerical simulations based on tDMRG algorithms [43–46]. We simulated the dynamics in a spin chain with open boundary conditions; the boundary effects are mitigated by choosing initial states that are locally stationary. We consider two scenarios: in the first scenario two thermal states are prepared at temperatures $T_{\text{L}} = 10$ and $T_{\text{R}} = 1$ on the left half and on the right half of the spin chain, respectively, where we set $J = 1$. In particular, denoting

$\mathbf{H}_{\mathrm{L}} = \sum_{\ell=1}^{L/2-2} \mathbf{q}_{1,\ell}^{+}$ and $\mathbf{H}_{\mathrm{R}} = \sum_{\ell=L/2+1}^{L-2} \mathbf{q}_{1,\ell}^{+}$, with $\mathbf{q}_{1,\ell}^{+}$ given in Eq. (19), the initial state reads

$$\boldsymbol{\rho}(0) = \frac{e^{-\mathbf{H}_{\mathrm{L}}/T_{\mathrm{L}}}}{\mathrm{tr}[e^{-\mathbf{H}_{\mathrm{L}}/T_{\mathrm{L}}}]} \otimes \frac{e^{-\mathbf{H}_{\mathrm{R}}/T_{\mathrm{R}}}}{\mathrm{tr}[e^{-\mathbf{H}_{\mathrm{R}}/T_{\mathrm{R}}}]}. \tag{132}$$

It is prepared by performing imaginary-time evolution of the infinite-temperature state, using the ancilla tDMRG annealing technique [46, 47]. Once thermalised, the two halves of the system are coupled and evolved in real time with the full folded Hamiltonian. The boundary conditions for the GHD continuity equations read

$$n_{-}(p) = \frac{1}{1 + e^{E(p)/T_{\mathrm{L}} - w_{\mathrm{L}}}}, \qquad n_{+}(p) = \frac{1}{1 + e^{E(p)/T_{\mathrm{R}} - w_{\mathrm{R}}}}, \tag{133}$$

where the energy shifts $w_{\pm}$ are computed by solving Eqs (38) and (39) at zero staggered magnetisation $m_{\mathrm{st}}^{z} = 0$.

In the second scenario, the initial state is prepared so as to have a nonzero staggered magnetisation in the left half of the spin chain. It reads

$$\boldsymbol{\rho}(0) = \frac{e^{h_{\mathrm{L}}\mathbf{S}_{\mathrm{st}}^{z} - \mathbf{H}_{\mathrm{L}}/T_{\mathrm{L}}}}{\mathrm{tr}[e^{h_{\mathrm{L}}\mathbf{S}_{\mathrm{st}}^{z} - \mathbf{H}_{\mathrm{L}}/T_{\mathrm{L}}}]} \otimes \frac{e^{-\mathbf{H}_{\mathrm{R}}/T_{\mathrm{R}}}}{\mathrm{tr}[e^{-\mathbf{H}_{\mathrm{R}}/T_{\mathrm{R}}}]}, \tag{134}$$

where we have chosen $h_{L} = 4$. To obtain the new left-hand side boundary condition for the GHD continuity equations, one has to compute the staggered magnetisation per macrosite according to Eq. (37). This only affects $w_{\mathrm{L}}$, through Eqs (38) and (39). For more details on numerical simulations see Appendix B.

# 6 Conclusion

In this work we have considered the dual folded XXZ model in the thermodynamic limit. We have developed the thermodynamic Bethe Ansatz that enables description of generalised Gibbs ensembles constructed from the strictly local charges and a special quasilocal conservation law related to the number of domain walls in a particle configuration. We have studied thermal states in some detail, exhibiting the low- and high-temperature asymptotic expansions of the energy, the magnetisation, and the specific heat. We have then moved our attention to the time evolution of inhomogeneous states prepared so as to be locally equivalent to local macrostates on large space and time scales. Such systems can be successfully investigated within the framework of generalised hydrodynamics. We have adapted this theory to the folded XXZ model and applied it to the crucial setting in which two macrostates (e.g., thermal states) are joined together at time $t = 0$. We found that the profiles of the expectation values of charge densities and currents are discontinuous in the ballistic scaling limit. To the best of our knowledge, there are no global symmetries that would protect some charges from developing discontinuities, contrary to the analogous scenario in the XXZ model.

Besides solving the first-order GHD equations numerically, we have also obtained analytical predictions for the position of the discontinuity and for the macrostates that describe the expectation values of the local observables in its neighbourhoods. We have tested our theoretical predictions against extensive tDMRG simulations of time evolution. The agreement was always fair, the discrepancies being reasonably explainable as finite-size or finite-time effects, and as numerical inaccuracies.

We warn the reader that, despite being sufficient for our purposes, the thermodynamic Bethe Ansatz that we developed (and, in turn, the generalised hydrodynamic theory) is not complete, as we did not include all quasilocal charges that complete the characterisation of

the configuration space. Some of these deficiencies could be filled rather easily. However, the solution would nevertheless remain deficient, as it would not take into account the non-abelian integrability of the folded XXZ model. The interest in the latter structure comes from its striking effects on the dynamics at intermediate times, therefore this aspect deserves further investigations, especially in the presence of interactions. Finally, despite non-ballistic behaviour being manifest in the inhomogeneous setting, we only worked out generalised hydrodynamics at the first order (ballistic scaling limit). In this regard we wonder whether the folded XXZ model is simple enough to make a step forward in the development of higher-order generalised hydrodynamics in interacting integrable systems.

## Acknowledgements

KB thanks G. Misguich for valuable suggestions.

**Funding information.** This work was supported by the European Research Council under the Starting Grant No. 805252 LoCoMacro.

## A  Sommerfeld expansion

In this section we provide more details about the Sommerfeld expansion in the folded XXZ model. First of all, we provide a bound for the error of approximation (51). To that aim, we note that all derivatives $\arcsin^{(n)} x$ of $\arcsin x$ are non-negative in the interval $[0,1]$. In addition, they are absolutely continuous in $[0, 1 - z_0]$, for any $0 < z_0 < 1$. Thus, for given $x \in [0, 1 - z - z_0]$ and $0 < z < 1 - z_0$ there exist real numbers $0 < \nu_0, \nu_1 < 1$, such that

$$
\begin{aligned}
\arcsin(z + x) - \sum_{n=0}^{j} \frac{\arcsin^{(n)}(z)}{n!} x^n &= \frac{\arcsin^{(j+1)}(z + \nu_0 x) x^{j+1}}{(j+1)!}, \\
\frac{1}{\sqrt{1 - (z + x)^2}} - \sum_{n=0}^{j} \frac{\arcsin^{(n+1)}(z)}{n!} x^n &= \frac{\arcsin^{(j+2)}(z + \nu_1 x) x^{j+1}}{(j+1)!}.
\end{aligned}
\tag{135}
$$

Expressions on the right-hand side are the Lagrange remainders of the Taylor polynomials. For some $0 < \nu_2 < 1$ we thus have

$$
\begin{aligned}
\int_0^{1-z} \frac{dx \log(1+e^{-4J\beta x})}{\pi\sqrt{1-(x+z)^2}} &= \int_{1-z-z_0}^{1-z} \frac{dx \log(1+e^{-4J\beta x})}{\pi\sqrt{1-(x+z)^2}} + \int_0^{1-z-z_0} \frac{dx \log(1+e^{-4J\beta x})}{\pi\sqrt{1-(x+z)^2}} \\
&= \log\left(1+e^{-4J\beta(1-z-\nu_2 z_0)}\right)\left[\frac{1}{2} - \frac{1}{\pi}\arcsin(1-z_0)\right] + \\
&\quad + \int_0^{1-z-z_0} \frac{dx \log(1+e^{-4J\beta x})}{\pi} \frac{\arcsin^{(j+2)}(z+\nu_1(x)x)x^{j+1}}{(j+1)!} + \\
&\quad + \sum_{n=0}^{j} \frac{\arcsin^{(n+1)}(z)}{n!} \int_0^{1-z-z_0} \frac{dx}{\pi} x^n \log(1+e^{-4J\beta x}) \\
&= \log\left(1+e^{-4J\beta(1-z-\nu_2 z_0)}\right)\left[\frac{1}{2} - \frac{1}{\pi}\arcsin(1-z_0)\right] - \\
&\quad - \sum_{n=0}^{j} \frac{\arcsin^{(n+1)}(z)}{n!} \int_{1-z-z_0}^{\infty} \frac{dx}{\pi} x^n \log(1+e^{-4J\beta x}) + \\
&\quad + \int_0^{1-z-z_0} \frac{dx \log(1+e^{-4J\beta x})}{\pi} \frac{\arcsin^{(j+2)}(z+\nu_1(x)x)x^{j+1}}{(j+1)!} + \\
&\quad + \sum_{n=0}^{j} \frac{\arcsin^{(n+1)}(z)}{n!} \int_0^{\infty} \frac{dx}{\pi} x^n \log(1+e^{-4J\beta x}),
\end{aligned}
\tag{136}
$$

where, in the second equality, we used Eq. (135) to rewrite the integral over the interval $[0, 1-z-z_0]$. The first two terms can easily be bounded with terms that are exponentially small in $\beta$. In particular, we have

$$
\begin{aligned}
\log\left(1+e^{-4J\beta(1-z-\nu_2 z_0)}\right)&\left[\frac{1}{2} - \frac{1}{\pi}\arcsin(1-z_0)\right] \leq \frac{1}{2}e^{-4J\beta(1-z-z_0)}, \\
\int_{1-z-z_0}^{\infty} \frac{dx}{\pi} x^n \log(1+e^{-4J\beta x}) &\leq \frac{\Gamma(n+1, 4J\beta(1-z-z_0))}{\pi(4J\beta)^{n+1}} \sim \frac{1}{4J\beta}e^{-4J\beta(1-z-z_0)}.
\end{aligned}
\tag{137}
$$

The last but one term in Eq. (136) is instead a power law correction, namely,

$$
\int_0^{1-z-z_0} \frac{dx \log(1+e^{-4J\beta x})}{\pi} \frac{\arcsin^{(j+2)}(z+\nu_1(x)x)x^{j+1}}{(j+1)!} \leq \frac{\arcsin^{(j+2)}(1-z_0)}{\pi(4J\beta)^{j+2}}(1-2^{-2-j})\zeta(j+3),
\tag{138}
$$

where $\zeta$ is the Riemann zeta function. For any $0 < z_0 < 1-z$ all of the correction terms are bounded by a power law. We therefore find the asymptotic expansion

$$
\int_0^{1-z} \frac{dx \log(1+e^{-4J\beta x})}{\pi\sqrt{1-[x+z]^2}} = \sum_{n=0}^{j} \frac{\arcsin^{(n+1)}(z)(1-2^{-n-1})\zeta(n+2)}{\pi(4J\beta)^{n+1}} + \mathcal{O}((J\beta)^{-j-2}),
\tag{139}
$$

where the integral in the last term of Eq. (136) has been worked out.

An analogous calculation allows us to work out the asymptotic expansion of expressions of the type $\langle f(\cos p)\rangle/(2\pi\rho_t) = \int \frac{dp}{2\pi} n(p) f(\cos p)$, for a given smooth function $f$. Indeed, we

have

$$\frac{\langle f(\cos p)\rangle}{2\pi\rho_t} = \int_{-\pi}^{\pi} \frac{dp}{2\pi} \frac{f(\cos p)}{1+e^{4J\beta[\cos p-\cos k(\beta)]}} = \int_{-1}^{1} \frac{dx}{\pi\sqrt{1-x^2}} \frac{f(x)}{1+e^{4J\beta[x-\cos k(\beta)]}} =$$

$$= \frac{1}{4J\beta} \int_0^{4J\beta[1+\cos k(\beta)]} \frac{f(\cos k(\beta)-\frac{y}{4J\beta})dy}{\pi\sqrt{1-[\cos k(\beta)-\frac{y}{4J\beta}]^2}} -$$

$$-\frac{1}{4J\beta} \int_0^{4J\beta[1+\cos k(\beta)]} \frac{f(\cos k(\beta)-\frac{y}{4J\beta})}{\pi\sqrt{1-[\cos k(\beta)-\frac{y}{4J\beta}]^2}} \frac{dy}{1+e^y} + \tag{140}$$

$$+\frac{1}{4J\beta} \int_0^{4J\beta[1-\cos k(\beta)]} \frac{f(\frac{y}{4J\beta}+\cos k(\beta))}{\pi\sqrt{1-[\frac{y}{4J\beta}+\cos k(\beta)]^2}} \frac{dy}{1+e^y} \approx$$

$$\approx g(\cos k(\beta)) + \sum_{n=0}^{\infty} \frac{(2-2^{-2n})\zeta(2n+2)}{(4J\beta)^{2n+2}} g^{(2n+2)}(\cos k(\beta)),$$

where

$$g(x) = \int_{-1}^{x} \frac{dy}{\pi} \frac{f(y)}{\sqrt{1-y^2}}, \tag{141}$$

and the last step is exponentially accurate in $\beta$. The lowest-order coefficients of the expansion of $g(x)$ about $x = \cos k_F$, for $f(x) = 1$ and $f(x) = x$, are shown in Table 1.

Table 1: Coefficients of the expansion of $g(x)$ around $\cos k_F$, for $f(x) = 1$ and $f(x) = x$, up to fourth order.

| $f(x)$ | $g(\cos k_F)$ | $g^{(1)}(\cos k_F)$ | $g^{(2)}(\cos k_F)$ | $g^{(3)}(\cos k_F)$ | $g^{(4)}(\cos k_F)$ |
|---|---|---|---|---|---|
| $1$ | $1-\frac{k_F}{\pi}$ | $\frac{1}{\pi\sin k_F}$ | $\frac{\cos k_F}{\pi\sin^3 k_F}$ | $\frac{2+\cos(2k_F)}{\pi\sin^5 k_F}$ | $\frac{3\cos k_F(4+\cos(2k_F))}{\pi\sin^7 k_F}$ |
| $x$ | $-\frac{\sin k_F}{\pi}$ | $\frac{1}{\pi\tan k_F}$ | $\frac{1}{\pi\sin^3 k_F}$ | $\frac{3\cos k_F}{\pi\sin^5 k_F}$ | $\frac{3(3+2\cos(2k_F))}{\pi\sin^7 k_F}$ |

In the main text we, in addition, showed

$$\cos k(\beta) = \cos k_F + \frac{\pi^2\cos k_F}{6\sin^2 k_F}\frac{1}{(4J\beta)^2} + \frac{\pi^4\cos k_F(43+9\cos(2k_F))}{720\sin^6 k_F}\frac{1}{(4J\beta)^4} + \mathcal{O}((J\beta)^{-6}). \tag{142}$$

Thus, at the fourth order, we have

$$\frac{\langle 1\rangle}{2\pi\rho^t} = 2 - \frac{\tan k_F}{\pi} + \frac{\pi}{3}\frac{\cos k_F}{\sin^3 k_F}(4J\beta)^{-2} + \frac{\pi^3\cos k_F(64+19\cos(2k_F))}{180\sin^7 k_F}(4J\beta)^{-4} + \mathcal{O}((4J\beta)^{-6}), \tag{143}$$

from which we obtain

$$\xi^{-1} = \frac{4\pi\cos k_F - 2\sin k_F}{\sin k_F} + \frac{4\pi^3\cos^3 k_F}{3\sin^5 k_F(4J\beta)^2} + \frac{\pi^5\cos^3 k_F(74+29\cos(2k_F))}{45\sin^9 k_F(4J\beta)^4} + \mathcal{O}((4J\beta)^{-6}), \tag{144}$$

and

$$2\pi\rho_t = \frac{2\pi\cos k_F}{\sin k_F} + \frac{2\pi^3\cos^3 k_F}{3\sin^5 k_F}(4J\beta)^{-2} + \frac{\pi^5\cos^3 k_F(74+29\cos(2k_F))}{90\sin^9 k_F}(4J\beta)^{-4} + \mathcal{O}((4J\beta)^{-6}). \tag{145}$$

Analogously, we find

$$\frac{\langle\cos p\rangle}{2\pi\rho_t} = -\frac{\sin k_F}{\pi} + \frac{\pi(3+\cos(2k_F))}{12\sin^3 k_F(4J\beta)^2} + \frac{\pi^3(739+580\cos(2k_F)+9\cos(4k_F))}{2880\sin^7(k_F)(4J\beta)^4} + \mathcal{O}((J\beta)^{-6}), \tag{146}$$

from which we can extract the first order of the low-temperature expansion of the energy

$$4J\langle\cos p\rangle = -8J\cos k_{\mathrm{F}} + \frac{4\pi^2 J\cos k_{\mathrm{F}}}{3\sin^2 k_{\mathrm{F}}(4J\beta)^2} + \frac{\pi^4 J\cos k_{\mathrm{F}}(43 + 29\cos(2k_{\mathrm{F}}))}{30\sin^6 k_{\mathrm{F}}(4J\beta)^4} + \mathcal{O}((J\beta)^{-6})\,. \tag{147}$$

# B  DMRG simulations

Numerical calculations are performed using C++ ITensor library [48]. We use the ancilla-based DMRG technique [46,47] to prepare initial state with a given inverse temperature $\beta$. We fix the maximal bond dimension to be $\mathcal{N}_{\mathrm{max}} = 600$ and a truncation, such that the sum of the discarded Schmidt values is smaller than $10^{-13}$. The imaginary-time step in the preparation of the initial state and the real-time step in the time evolution of the state are both set to $\delta t = \delta\beta = 0.01 J^{-1}$. The code and the animated evolution of the energy profile are freely accessible (see also Fig. 9).[14,15,16]

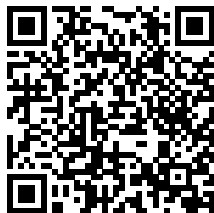
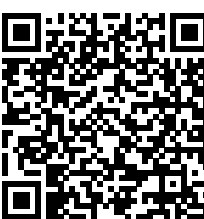

Figure 9: QR codes that link to animated version of the Fig. 8. Alternatively, follow the links in the footnotes below.

## B.1  Preparing the initial state

In order to use algorithms based on the matrix product state representation, the density matrix is vectorised as

$$|\psi(\beta)\rangle = e^{-\frac{\beta}{2}\mathbf{H}}|\psi(0)\rangle\,, \tag{148}$$

where the infinite temperature state is defined as a product of maximally entangled physical and ancilla spins:

$$|\psi(0)\rangle = \bigotimes_j \frac{1}{\sqrt{2}}(|\uparrow\downarrow\rangle_{j,\tilde{j}} - |\downarrow\uparrow\rangle_{j,\tilde{j}})\,. \tag{149}$$

Here, indices $j$ and $\tilde{j}$ denote the physical and the ancilla spin, respectively.

In numerical simulations we consider the Hamiltonian (8) with open boundary conditions. It is written as a sum of three parts $\mathbf{H} = \mathbf{H}_1 + \mathbf{H}_2 + \mathbf{H}_3$, such that local terms in each mutually commute. This allows to approximate the evolution operator by Trotter gates [49–51]. In the present work we use second order Trotter-Suzuki approximation

$$e^{-\delta t(\mathbf{H}_1 + \mathbf{H}_2 + \mathbf{H}_3)} = e^{-\frac{\delta t}{2}\mathbf{H}_1}e^{-\frac{\delta t}{2}\mathbf{H}_2}e^{-\delta t\mathbf{H}_3}e^{-\frac{\delta t}{2}\mathbf{H}_2}e^{-\frac{\delta t}{2}\mathbf{H}_1} + \mathcal{O}(\delta t^3)\,, \tag{150}$$

which can be factorised in terms of Trotter gates $\exp\left(i\frac{J}{2}\delta t(\sigma_\ell^x\sigma_{\ell+2}^x + \sigma_\ell^y\sigma_{\ell+2}^y)(1 - \sigma_{\ell+1}^z)\right)$ that act on the physical sites only, the ancilla spins remaining untouched. This helps avoid additional truncation.

---

[14]The code: https://github.com/kbidzhiev/Folded_XXZ

[15]Evolution of the energy profile: https://github.com/kbidzhiev/Folded_XXZ/raw/master/Pictures/Energy_profile.gif

[16]Evolution of the rescaled energy profile: https://github.com/kbidzhiev/Folded_XXZ/raw/master/Pictures/Energy_profile_rescaled.gif

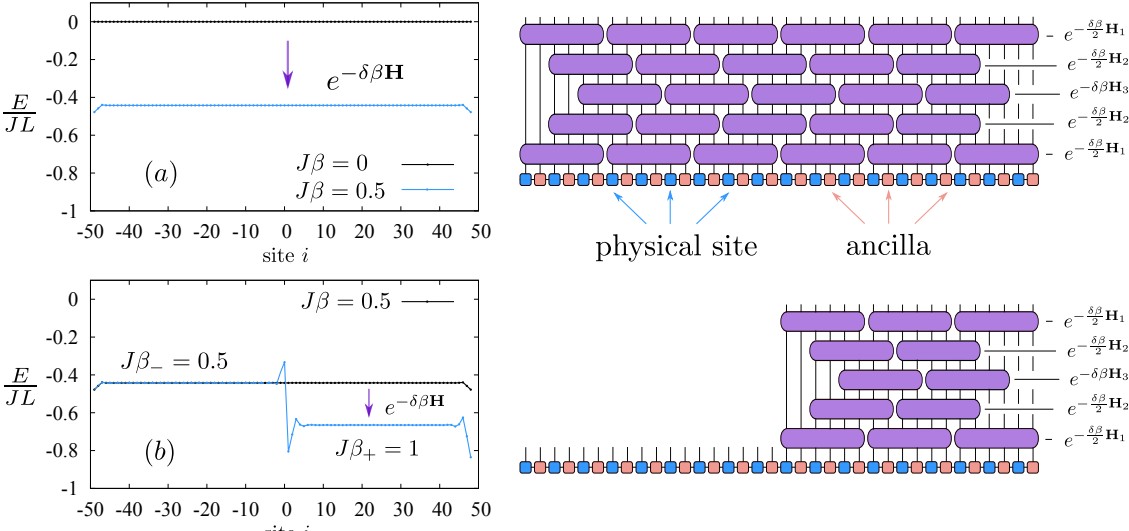

Figure 10: Panel (a) shows the energy profile for different values of $\beta$ (left-hand side) and a cartoon of the cooling procedure (right-hand side). Trotter gates act on the wavefunction and gradually decrease the temperature from $T = \infty$ ($\beta = 0$) to $T_L$ ($\beta_-$). Panel (b) shows the energy profile for a state with temperature $T_L$ ($T_R$) on the left (right) half of the system. We keep acting with Trotter gates on the right-hand side, until the temperature $T_R$ is reached.

Starting from the infinite temperature state $|\psi(0)\rangle$, we gradually cool the system by imaginary-time evolution until it reaches a state $|\psi(\beta_-)\rangle$, with $\beta_- = 1/T_L$. For a typical example, see panel (a) of Fig. 10. The further cooling is provided by Trotter gates that act only on the right half of the system, until the right subsystem reaches the temperature $\beta_+ = 1/T_R$, as shown in panel (b) of Fig. 10. This protocol creates initial correlations between the left and the right subsystems. Alternatively, one could zero the initial correlations between the two parts by acting with the Trotter gates on both subsystems separately, omitting the three central Trotter gates that overlap with the junction between the two subsystems. The latter protocol, however, introduces larger oscillatory finite-size effects. The energy profile that results from this last protocol is presented in panel (a) of Fig. 8.

## B.2 Energy density

At time $t = 0$ the two halves of the system are prepared at different temperatures $T_L = 10J$ and $T_R = J$, then left to evolve in time with the Hamiltonian of the full system with open boundary conditions. In the vicinity of the centre and boundaries one can observe Friedel-like oscillations, which increase as the temperature drops to zero. The oscillations reach the maximal amplitude in the ground state, as shown in panel (b) of Fig. 11. They decay as the inverse square root of the "chord length", i.e.,

$$r = 1/\sqrt{\frac{L+1}{\pi} \sin\left(\frac{\pi x}{L+1}\right)} \xrightarrow{L \to \infty} \frac{1}{\sqrt{x}}, \tag{151}$$

where $x$ denotes the distance from the boundary of the system, as measured in the number of spin sites.

The energy front in the left (right) half of the system propagates with velocity $|\zeta_{lc}^{\pm}| \approx 3$, measured in macrosites, or $\approx 6$, measured in terms of spin sites. This gives rise to a "light cone" shown in Figs 7 and 8. The energy smoothly tends to the initial-state values near the

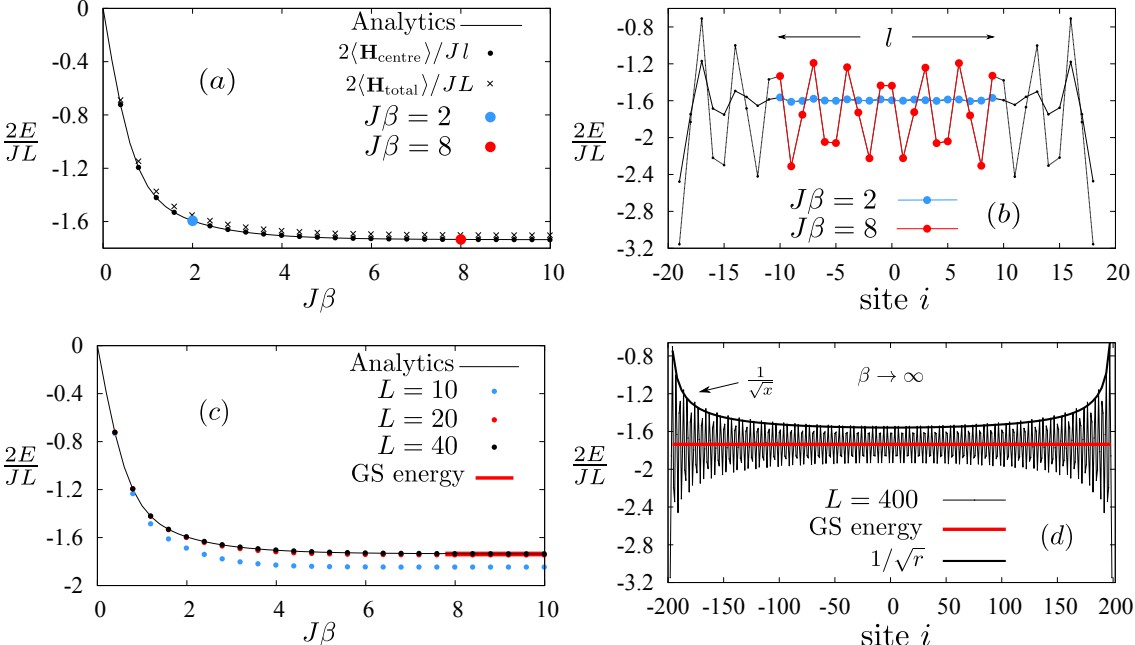

Figure 11: Panel (a) shows the energy density as a function of $J\beta$. Discrepancy between the analytical and numerical result for the energy $\langle \mathbf{H}_{total} \rangle$ is a consequence of the boundary Friedel-like oscillations. To avoid the inaccuracies due to these oscillations, we average the energy in the vicinity of the centre, i.e., on sites $-L/4 \leq i < L/4$, as shown in panel (b) (the corresponding average is denoted by $\langle \mathbf{H}_{centre} \rangle$). The effect of oscillations is larger at low temperatures and reaches the maximum in the ground state – see panel (d). Panel (c) shows the energy density for different system sizes. Already for relatively small systems of $\sim 40$ spins the results perfectly agree with the analytical predicitions. Panel (d) shows the energy profile in the ground state ($\beta \rightarrow \infty$). The Friedel-like oscillations reach their maximal value and slowly vanish as $\sim 1/\sqrt{x}$, where $x$ is the distance from the boundary. The envelope of the oscillations is described by the "chord length" formula.

light cone edges. The discontinuity in the numerically obtained profile is relatively small on the achievable time scales.

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
