# Peer review of "The Folded Spin-1/2 XXZ Model: II. Thermodynamics and Hydrodynamics with a Minimal Set of Charges"

_SciPost Physics, doi:SciPost Phys. 10, 099 (2021)_

## Round 2 · Referee Report · Anonymous (Referee 1) · 2021-1-10

Strengths

1 - The overall motivation for the work, as well as main results of the manuscript, are well explained and sound.

2 - Proper treatment of strong coupling regimes is often difficult and subject to uncontrolled approximations

Report

The manuscript by Zadnik et al studies properties of the folded spin-1/2 XXZ model. It is a continuation of the initial work by two authors of the present manuscript to construct an effective Hamiltonian in the strong coupling limit of the model. In this work, the authors focus on the thermodynamic Bethe ansatz description of the model and on the generalized hydrodynamics description. Finally, they consider some specific examples in nonequilibrium dynamics where two semi-infinite chains are put into contact.

The overall motivation for the work, as well as main results of the manuscript, are well explained and sound. I understand that a proper treatment of strong coupling regimes is often difficult and subject to uncontrolled approximations. I am, however, not an expert in neither the Bethe ansatz technique nor in the generalized hydrodynamics. Still, I find the contribution of the authors as an interesting new development in the field of the well-studied spin-1/2 XXZ model and therefore recommend the manuscript for publication in SciPost Physics.
  • validity: high
  • significance: high
  • originality: high
  • clarity: high
  • formatting: excellent
  • grammar: excellent

Author:  Lenart Zadnik  on 2021-02-10  [id 1222]

(in reply to Report 1 on 2021-01-10)

Dear Referee (2021-1-10),

We appreciate your positive report on our manuscript.

Yours sincerely,

The Authors

---

## Round 2 · Referee Report · Anonymous (Referee 2) · 2021-2-1

Strengths

1- the paper is well written and present enough details about this new model; 2- the folded XXZ provides a nice example of a spin chain which is interacting with a simple but rich Bethe-Ansatz structure; 3- Comparison between numerics and analytics is shown with very good agreement; 4- Non-trivial discontinuous behavior identified in the stationary state emerging from the partitioning protocol.

Weaknesses

1- it is not reported how well the prediction of the folded XXZ would compare with the full XXZ at large anisotropy.

Report

In this work, the authors continue their study of the so-called "folded XXZ model" which emerges by considering an appropriate limit of the large anisotropy Heisenberg spin chain. The computation of the thermodynamic bethe ansatz and the generalised hydrodynamics (GHD) equation provide interesting results for both the equilibrium and out-of-equilibrium properties of the model.

As an application the authors consider the standard partitioning protocol, obtained joining two thermal states. In this case, a solution of the GHD equation is given in the ballistic limit. Interestingly enough, quite generically the profile of local observables shows a discontinuity. Such a property even appears in the Yang-Yang entropy, which gives the stationary limit of finite interval entanglement entropy.

I believe that this is a nice and interesting work. Out-of-equilibrium properties of integrable systems have attracted tremendous interest in recent years and the authors have identified a nice model for which most of the Bethe-Ansatz technicalities have a rather simple form while still remain highly non-trivial.

Requested changes

1- In the introduction, could the author explain the difference between what they dub "folded picture" and the standard "interaction picture"? 2- Could it be sketched at least how Eq. 5 follows from Eq. 4? 3- I had some troubles understanding the notation introduced after Eq. 9 for the positions of the down spins, i.e. $2 \ell_j' + b_j$. It should be stated more clearly that $b_j \in {0,1}$, i.e. they are the parity $\mod(2)$ of the position index. Please state it more explicitly. 4- Could it be clarified why Eq. 109 is only a conjecture? What is the technical difficulty with the staggered magnetization? 5- There seems to be a problem with the caption in Fig. 8, where the labels (b) and (c) are exchanged.

  • validity: top
  • significance: good
  • originality: good
  • clarity: good
  • formatting: good
  • grammar: excellent

Author:  Lenart Zadnik  on 2021-02-10  [id 1221]

(in reply to Report 2 on 2021-02-01)

Dear Referee (2021-2-1),

Thank you for your positive report and helpful comments on the manuscript. We have incorporated the requested changes in the new version of the paper. Below we address them point by point:

(1) The folded formulation's main distinction from the standard interaction picture lies in the fact that it is the state rather than the operator that evolves with a time-independent Hamiltonian; furthermore, both state and operators undergo an additional (stationary) unitary transformation. A comment has been added at the end of the paragraph below Eq. (3).

(2) The idea behind the derivation of the effective folded Hamiltonian is very simple: the time evolution with the full Heisenberg Hamiltonian is split as $e^{-i H t}=e^{-i H_I t}U(t;\kappa)$. Then, $U(t;\kappa)$ is expanded in the inverse coupling constant $\kappa$; the leading order reads $e^{-i H_F t}$. This clarification has been incorporated in the form of a footnote.

(3) Appropriate changes have been made to the corresponding paragraph, below Eq. (9).

(4) The charges that pertain to the configuration degrees of freedom, namely $(b_1,\ldots,b_N)$, are not known, except for the staggered magnetisation. Their expectation values are not fixed by the momenta, which makes the derivation of the corresponding hydrodynamic equations out of scope. On the other hand, since in the TD limit the only relevant parameter referring to the configuration is $\mu$, we expect and conjecture all configuration-related degrees of freedom to behave in a similar fashion.

(5) The caption has been corrected, thank you for noticing.

We express our appreciation for your comments again.

Yours sincerely,

The Authors

---

## Round 3 · Author Response

Dear Editor,
Thank you for sharing the referee reports, as well as for the Editorial recommendation. Per your request we have incorporated minor changes proposed in the recent referee report. For your convenience the changes are listed below.
Additional explanations and remarks on the referee reports are visible in the direct replies to their reports.
Yours sincerely,
The Authors
Thank you for sharing the referee reports, as well as for the Editorial recommendation. Per your request we have incorporated minor changes proposed in the recent referee report. For your convenience the changes are listed below.
Additional explanations and remarks on the referee reports are visible in the direct replies to their reports.
Yours sincerely,
The Authors

---

## Round 3 · List of Changes

(1) A short comment on the difference between the interaction and folded picture has been added at the end of the paragraph below Eq. (3).
(2) A very brief sketch of how to obtain the effective Hamiltonian has been added in a footnote.
(3) The indices that denote the spin up positions have been appropriately explained after Eq. (9).
(4) The caption of Fig. 8 has been corrected.
(2) A very brief sketch of how to obtain the effective Hamiltonian has been added in a footnote.
(3) The indices that denote the spin up positions have been appropriately explained after Eq. (9).
(4) The caption of Fig. 8 has been corrected.

---

## Editorial Decision

published